# Representation Ensembling for Synergistic Lifelong Learning with Quasilinear Complexity

## Abstract

In biological learning, data are used to improve performance not only on the current task, but also on previously encountered, and as yet unencountered tasks. In contrast, classical machine learning which we define as starting from a blank slate, or *tabula rasa*, using data only for the single task at hand. While typical transfer learning algorithms can improve performance on future tasks, their performance on prior tasks degrades upon learning new tasks (called forgetting). Many recent approaches for continual or lifelong learning have attempted to *maintain* performance given new tasks. But striving to avoid forgetting sets the goal unnecessarily low: the goal of lifelong learning, whether biological or artificial, should be to improve performance on both past tasks (backward transfer) and future tasks (forward transfer) with any new data. Our key insight is that even though learners trained on other tasks often cannot make useful decisions on the current task (the two tasks may have non-overlapping classes, for example), they may have learned *representations* that are useful for this task. Thus, although ensembling decisions is not possible, ensembling representations can be beneficial whenever the distributions across tasks are sufficiently similar. Moreover, we can ensemble representations learned independently across tasks in quasilinear space and time. We therefore propose two algorithms: representation ensembles of (1) trees and (2) networks. Both algorithms demonstrate forward and backward transfer in a variety of simulated and real data scenarios, including tabular, image, and spoken, and adversarial tasks. This is in stark contrast to the reference algorithms we compared to, most of which failed to transfer either forward or backward, or both, despite that many of them require quadratic space or time complexity.

## 1 Introduction

Learning is the process by which an intelligent system improves performance on a given task by leveraging data (Mitchell, 1999). In biological learning, learning is lifelong, with agents continually building on past knowledge and experiences, improving on many tasks given data associated with any task. For example, learning a second language often improves performance in an individual's native language (Zhao et al., 2016). In classical machine learning, the system often starts with essentially zero knowledge, a "tabula rasa", and is optimized for a single task (Vapnik & Chervonenkis, 1971; Valiant, 1984). While it is relatively easy to *simultaneously* optimize for multiple tasks (multi-task learning) (Caruana, 1997), it has proven much more difficult to *sequentially* optimize for multiple tasks (Thrun, 1996; Thrun & Pratt, 2012). Specifically, classical machine learning systems, and natural extensions thereof, exhibit "catastrophic forgetting" when trained sequentially, meaning their performance on the prior tasks drops precipitously upon training on new tasks (McCloskey & Cohen, 1989; McClelland et al., 1995). This is in contrast to many biological learning settings, such as the second language learning setting mentioned above.

In the past 30 years, a number of sequential task learning algorithms have attempted to overcome catastrophic forgetting. These approaches naturally fall into one of two camps. In one, the algorithm has fixed resources, and so must reallocate resources (essentially compressing representations) in order to incorporate new knowledge (Kirkpatrick et al., 2017; Zenke et al., 2017; Li & Hoiem, 2017; Schwarz et al., 2018; Finn et al., 2019). Biologically, this corresponds to adulthood, where brains have a nearly fixed or decreasing

number of cells and synapses. In the other, the algorithm adds (or builds) resources as new data arrive (essentially ensembling representations) (Ruvolo & Eaton, 2013; Rusu et al., 2016; Lee et al., 2019). Biologically, this corresponds to development, where brains grow by adding cells, synapses, etc. A close resemblance to this can be found in Sodhani et al. (2020) where the model adaptively expands the capacity when the capacity of the model saturates.

Approaches from both camps demonstrate some degree of continual (or lifelong) learning (Parisi et al., 2019). In particular, they can sometimes learn new tasks while not catastrophically forgetting old tasks. However, as we will show, many state of the art lifelong learning algorithms are unable to transfer knowledge forward, and most of them are able to transfer knowledge backward with small sample sizes where it is particularly important. This inability to synergistically learn has been identified as one of the key obstacles limiting the capabilities of artificial intelligence (Pearl, 2019; Marcus & Davis, 2019).

Our work falls into the resource growing camp in which each new task is learned with additional representational capacity. Our key innovation is the introduction of ensembling independent representations, rather than ensembling decisions (as in random forests (Breiman, 2001) or network ensembles (Pinto et al., 2009)). This is in contrast to ensembling representations that are conditionally dependent on the past representations (like gradient boosting trees (Chen & Guestrin, 2016) and ProgNN (Rusu et al., 2016)) or both past and future representations (DF-CNN (Lee et al., 2019) and all of the fixed resource algorithms). By virtue of learning them independently, we overcome interference from the past and the future representations to the current representation, and thereby, mitigate catastrophic forgetting. Moreover, the representations interact with each other through a channel layer which enables both forward and backward transfer. In doing so, we also reduce computational time and space from quadratic to quasilinear (i.e., linear up to polylog terms).

In this paper, we consider a simplified learning environment similar to the ones used in Kirkpatrick et al. (2017); Schwarz et al. (2018); Zenke et al. (2017); Li & Hoiem (2017); Rusu et al. (2016); Lee et al. (2019) where we know the task identities and the data arrives in batches rather than in a streaming fashion. Moreover, we keep the data from the previous tasks for achieving backward transfer between the tasks. Robins (1995); Shin et al. (2017); van de Ven et al. (2020) showed if one algorithm is allowed to keep the old task data, one can mitigate catastrophic forgetting. However, it is not obvious how one can improve from the future tasks, i.e., achieve backward transfer. Again, the problem of data storage can be effectively solved by learning a generative model simultaneously while learning the represenations for the tasks. Therefore, throughout the paper, we mainly focus on how synergistic learning from past and future tasks can be achieved by synergistically combining independent representations over the tasks in a simplified environment. In this work, we introduce two complementary synergistic learning algorithms, one based on decision forests (Syngeristic Forests, SᴠɴF), and another based on deep networks (Synergistic Networks, SᴠɴN). Both SᴠɴF and SᴠɴN demonstrate forward and backward transfer, while maintaining computational efficiency. Simulations illustrate their learning capabilities, including performance properties in the presence of adversarial tasks. We then demonstrate their learning capabilities in vision and language benchmark applications. Although the algorithms presented here are primarily resource building, we illustrate that they can effectively leverage prior representations. This ability implies that the algorithm can convert from a "juvenile" resource building state to the "adult" resource recruiting state – all while maintaining key synergistic learning capabilities and efficiencies.

## 2 Background

### 2.1 Classical Machine Learning

Classical supervised learning (Mohri et al., 2018) considers random variables $(X, Y) \sim P_{X,Y}$, where $X$ is an $\mathcal{X}$-valued input, $Y$ is a $\mathcal{Y}$-valued label (or response), and $P_{X,Y} \in \mathcal{P}_{X,Y}$ is the joint distribution of $(X, Y)$. Given a loss function $\ell : \mathcal{Y} \times \mathcal{Y} \to [0, \infty)$, the goal is to find the hypothesis (also called predictor), $h : \mathcal{X} \to \mathcal{Y}$ that minimizes expected loss, or *risk*, $R(h) = \mathbb{E}_{X,Y} [\ell(h(X), Y)]$. A learning algorithm is a function $f$ that maps data sets ($n$ training samples) to a hypothesis, where a data set $\mathbf{S}_n = \{X_i, Y_i\}_{i=1}^n$ is a set of $n$ input/response pairs. Assume $n$ samples of $(X, Y)$ pairs are independently and identically distributed from some true but unknown $P_{X,Y}$ (Mohri et al., 2018). A learning algorithm is evaluated on its generalization

error (or expected risk): $\mathbb{E}\left[R(f(\mathbf{S}_n))\right]$, where the expectation is taken with respect to the true but unknown distribution governing the data, $P_{X,Y}$. The goal is to choose a learner $f$ that learns a hypothesis $h$ that has a small generalization error for the given task (Bickel & Doksum, 2015).

## 2.2 Lifelong Learning

Lifelong supervised learning generalizes classical supervised machine learning in a few ways: (i) instead of one task, there is an environment $\mathcal{T}$ of (possibly infinitely) many tasks drawn according to some distribution $P_t$, (ii) data-label pair $(X, Y)$ for each task sampled from some distribution $P_{s_t}$ arrive sequentially, rather than in batch mode, and (iii) there are computational complexity constraints on the learning algorithm and hypotheses. One can consider the risk in the supervised lifelong learning setting as:

$$R(h_n) = \mathbb{E}_{t \sim P_t}[\mathbb{E}_{(X,Y) \sim P_{s_t}}[\ell(h_n(X), Y)]. \tag{1}$$

Implicit in the above equation is that we are integrating with respect to the task distribution, $P_t$ in the outer expectation, and therefore concerned not just with past tasks, but also all possible future tasks. That said, we are not explicitly minimizing the risk in equation 1 in our proposed approach. The computational complexity constraints for lifelong learning is crucial, though often implicit. For example, consider the algorithm that stores all the data, and then retrains everything from scratch each time a new sample arrives. Without computational constraints, such an algorithm could be classified as a lifelong learner; we do not think such a label is appropriate for that algorithm. Thus, we only consider learners $h_n$ lifelong learners assuming their performance scales sub-quadratically with sample size (see below for details). Note that equation 1 says a significant improvement on one task (or a set of tasks) should not come at a significant loss of performance on the other tasks. The risk defined in equation 1 allows the learner to lose performance on a subset of the tasks while gaining in performance on other tasks. The goal in lifelong learning therefore is, given new data and a new task, use all the existing data to achieve lower generalization error (or expected risk) on this new task, while also using the new data to obtain a lower generalization error on the previous tasks. This is distinct from classical online learning scenarios (Cesa-Bianchi & Lugosi, 2006), because the previously experienced tasks may recur, so we are concerned about maintaining and improving performance on those tasks as well. In "task-aware" scenarios, the learner is aware of all task details for all tasks, meaning that the hypotheses are of the form $h : \mathcal{X} \times \mathcal{T} \to \mathcal{Y}$. In "task-unaware" (or agnostic (Zeno et al., 2018)) scenarios the learner may not know that the task has changed at all, which means that the hypotheses are of the form $h : \mathcal{X} \to \mathcal{Y}$. We only address task-aware scenarios here.

## 2.3 Reference algorithms

We compared our approaches to 13 reference lifelong learning methods. These algorithms can be classified into two groups based on whether they add capacity resources per task, or not. Among them, ProgNN (Rusu et al., 2016) and Deconvolution-Factorized CNNs (DF-CNN) (Lee et al., 2019) learn new tasks by building new resources. For ProgNN, for each new task a new "column" of network is introduced. In addition to introducing this column, lateral connections from all previous columns to the new column are added. These lateral connections are computationally costly, as explained below. DF-CNN (Lee et al., 2019) is a lifelong learning algorithm that improves upon ProgNN by introducing a knowledge base with lateral connections to each new column, thereby avoiding all pairwise connections, and dramatically reducing computational costs. We also compare two variants of exact replay (Total Replay and Partial Replay) (Rolnick et al., 2019). Both store all the data they have ever seen, but Total Replay replays all of it upon acquiring a new task, whereas Partial Replay replays $M$ samples, randomly sampled from the entire corpus, whenever we acquire a new task with $M$ samples. We have also compared with more constrained ways of replaying old task data like- Model Zoo (Ramesh & Chaudhari, 2021), Averaged Gradient Episodic Memory (A-GEM) (Chaudhry et al., 2018), Experience Replay (ER) (Chaudhry et al., 2019) and Task-based Accumulated Gradients (TAG) (Malviya et al., 2021) for lifelong learning. Among them Model Zoo builds on our approach and ensembles multiple representations using the boosting approach. In Model Zoo, the total number of models within the ensemble (the number of episode) was capped at the total number of tasks to make it comparable with our approach. For A-GEM and ER, the size of episodic memory is set to store 1 example per class. On the other hand,

TAG stores the gradients or directions the model took while learning a specific task instead of storing past examples.

The other five algorithms, Elastic Weight Consolidation (EWC) (Kirkpatrick et al., 2017), Online-EWC (O-EWC) (Schwarz et al., 2018), Synaptic Intelligence (SI) (Zenke et al., 2017), Learning without Forgetting (LwF) (Li & Hoiem, 2017), and "None," all have fixed capacity resources. For the baseline "None", the network was incrementally trained on all tasks in the standard way while always only using the data from the current task. The implementations for all of the algorithms are adapted from open source codes (Lee et al., 2019; van de Ven & Tolias, 2019); for implementation details, see Appendix E.

## 3 Evaluation Criteria

Others have previously introduced criteria to evaluate transfer, including forward and backward transfer (Lopez-Paz & Ranzato, 2017; Benavides-Prado et al., 2018). These definitions typically compare the difference, rather than the ratio, between learning with and without transfer. Pearl Judea (2018) introduced the transfer benefit ratio, which builds directly off relative efficiency from classical statistics (Bickel & Doksum, 2015). Our definitions are closely related to this. *Learning efficiency* is the ratio of the generalization error of an algorithm that has learned on one dataset, as compared to the generalization error of that same algorithm on a different dataset. Typically, we are interested in situations where the former dataset is a subset of the latter dataset. Let $R^t$ be the risk associated with task $t$, and $\mathbf{S}_n^t$ be the data from $\mathbf{S}_n$ that is specifically associated with Task $t$ with sample size $n_t$, so $R^t(f(\mathbf{S}_n^t))$ is the risk on Task $t$ of the hypothesis learned by $f$ only using Task $t$ data, and $R^t(f(\mathbf{S}_n))$ denotes the risk on Task $t$ of the hypothesis learned on all the data. Note that, $\sum_{t=1}^T n_t = n$.

**Definition 1 (Learning Efficiency)** *The learning efficiency of algorithm $f$ for given Task $t$ with sample size $n$ is* $\mathsf{LE}_n^t(f) := \mathbb{E}\left[R^t\left(f(\mathbf{S}_n^t)\right)\right] / \mathbb{E}\left[R^t\left(f(\mathbf{S}_n)\right)\right]$. *We say that algorithm $f$ has learned all the tasks up to $t$ with data $\mathbf{S}_n$ if and only if $\mathsf{LE}_n^t(f) > 1$ for all the tasks up to $t$.*

To evaluate a lifelong learning algorithm while respecting the streaming nature of the tasks, it is convenient to consider two extensions of learning efficiency. *Forward* learning efficiency is the expected ratio of the risk of the learning algorithm with (i) access only to Task $t$ data, to (ii) access to the data up to and including the last observation from Task $t$. This quantity measures the relative effect of previously seen out-of-task data on the performance on Task $t$. Formally, let $N^t = \max\{i : T_i = t\}$, be the index of the last occurrence of task $t$ in the data sequence. Let $\mathbf{S}_n^{\leq t} = \{(X_1, Y_1, T_1), ..., (X_{N^t}, Y_{N^t}, T_{N^t})\}$ be all data up to and including that data point.

**Definition 2 (Forward Learning Efficiency)** *The forward learning efficiency of $f$ for task $t$ given $n$ samples is* $\mathsf{FLE}_n^t(f) := \mathbb{E}\left[R^t\left(f(\mathbf{S}_n^t)\right)\right] / \mathbb{E}\left[R^t\left(f(\mathbf{S}_n^{\leq t})\right)\right]$.

We say an algorithm (positive) forward transfers for task $t$ if and only if $\mathsf{FLE}_n^t(f) > 1$. In other words, if $\mathsf{FLE}_n^t(f) > 1$, then the algorithm has used data associated with past tasks to improve performance on task $t$.

One can also determine the rate of *backward* transfer by comparing $R^t\left(f(\mathbf{S}_n^{\leq t})\right)$ to the risk of the hypothesis learned having seen the entire training dataset. More formally, backward learning efficiency is the expected ratio of the risk of the learned hypothesis with (i) access to the data up to and including the last observation from task $t$, to (ii) access to the entire dataset. Thus, this quantity measures the relative effect of previous task data on the performance on Task $t$.

**Definition 3 (Backward Learning Efficiency)** *The backward learning efficiency of $f$ for Task $t$ given $n$ samples is* $\mathsf{BLE}_n^t(f) := \mathbb{E}\left[R^t\left(f(\mathbf{S}_n^{\leq t})\right)\right] / \mathbb{E}\left[R^t\left(f(\mathbf{S}_n)\right)\right]$.

We say an algorithm (positive) backward learns Task $t$ if and only if $\mathsf{BLE}_n^t(f) > 1$. In other words, if $\mathsf{BLE}_n^t(f) > 1$, then the algorithm has used data associated with future tasks to improve performance on previous tasks.

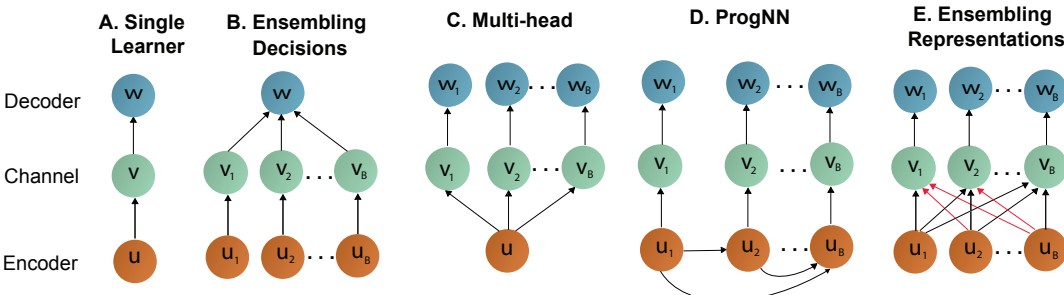

Figure 1: Schemas of composable hypotheses. Ensembling decisions (as output by the channels) is a well-established practice, including random forests and gradient boosted trees. Ensembling representations (learned by the encoders) was previously used in lifelong learning scenarios, but were not trained independently, thereby causing interference or forgetting. Note that the new encoders interact with the previous encoders through the channel layer (indicated by red arrows), thereby, enabling backward transfer. Again the old encoders interact with the future encoders (indicated by black arrows), thereby, enabling forward transfer.

After observing $m$ tasks, the extent to which the LE for the $j^{th}$ task comes from forward transfer versus from backward transfer depends on the order of the tasks. If we have a sequence in which tasks do not repeat, learning efficiency for the first task is all backward transfer, for the last task it is all forward transfer, and for the middle tasks it is a combination of the two. In general, LE factorizes into FLE and BLE:

$$\mathsf{LE}_n^t(f) = \frac{\mathbb{E}\left[R^t\left(f(\mathbf{S}_n^t)\right)\right]}{\mathbb{E}\left[R^t\left(f(\mathbf{S}_n)\right)\right]} = \frac{\mathbb{E}\left[R^t\left(f(\mathbf{S}_n^t)\right)\right]}{\mathbb{E}\left[R^t\left(f(\mathbf{S}_{\tilde{n}}^{\leq t})\right)\right]} \times \frac{\mathbb{E}\left[R^t\left(f(\mathbf{S}_{\tilde{n}}^{\leq t})\right)\right]}{\mathbb{E}\left[R^t\left(f(\mathbf{S}_n)\right)\right]}.$$

Throughout, we will report log LE so that positive learning corresponds to LE > 1. In a lifelong learning environment having $T$ tasks drawn with replacement from $\mathcal{T}$, learner $f$ $\boldsymbol{w}$-lifelong learns tasks $t \in \mathcal{T}$ if the log of the convex combination of learning efficiencies is greater than 0, that is,

$$\log \sum_{t \in \mathcal{T}} w_t \cdot \mathsf{LE}_n^t(f) > 0 \ . \tag{2}$$

Note that the hardest performance to achieve is when $w_t$ puts equal weights on every task. We say an agent has **synergistically learned** if the agent has positively learned for all tasks in all of the possible convex combinations of $w$. Again, we say an agent has **catastrophically forgotten**, if it has negative backward transfer for all the tasks. See Appendix A for a concrete example scenario of using our proposed metrics along with other metrics proposed in the literature.

## 4 Representation Ensembling Algorithms

Our approach to lifelong learning is based on decomposition of the hypothesis learned by a model into an encoder, a channel, and a decoder (Cover & Thomas, 2012; Cho et al., 2014) (Figure 1A): $h(\cdot) = w \circ v \circ u(\cdot)$. The encoder, $u : \mathcal{X} \mapsto \tilde{\mathcal{X}}$, maps an $\mathcal{X}$-valued input into an internal representation space $\tilde{\mathcal{X}}$ (Vaswani et al., 2017; Devlin et al., 2018). The channel $v : \tilde{\mathcal{X}} \mapsto \Delta_{\mathcal{Y}}$ maps the transformed data into a posterior distribution (or, more generally, a score). For example, consider we have a dataset partitioned into a training and a held-out set. Now we can learn a decision tree using the training data which will give us the encoder. Next, by pushing the held-out dataset through the tree, we can learn the channel, i.e., posteriors in the leaf-nodes. The channel thus gives scores for each data point denoting the probability of that data point belonging to a specific class. Finally, a decoder $w : \Delta_{\mathcal{Y}} \mapsto \mathcal{Y}$, produces a predicted label. See Appendix B for a detailed and concrete example using a decision tree.

One can generalize the above decomposition by allowing for multiple encoders. Given $B$ different encoders, one can attach a single channel to each encoder, yielding $B$ different channels (Figure 1B). Doing so requires

generalizing the definition of a decoder, which would operate on multiple channels. Such a decoder ensembles the *decisions*, because here each channel provides the final output based on the encoder. This is the learning paradigm behind boosting (Freund, 1995) and bagging (Breiman, 1996)—indeed, decision forests are a canonical example of a decision function operating on a collection of $B$ outputs (Breiman, 2001). A decision forest learns $B$ different decision trees, each of which has a tree structure corresponding to an encoder. Each tree is assigned a channel that outputs that single tree's guess as to the class of any probability that an observation is in any class. The decoder outputs the most likely class averaged over the trees.

Although the task specific structure in Figure 1B can provide useful decision on the corresponding task, they can not, in general, provide meaningful decisions on other tasks because those tasks might have completely different class labels, for example. Therefore, in the multi-head structure (Figure 1C) a single encoder is used to learn a joint representation from all the tasks and a separate channel is learned for each task to get the score or class conditional posteriors for each task which is followed by each task specific decider (Kirkpatrick et al., 2017; Schwarz et al., 2018; Zenke et al., 2017). Again, further modification of the multi-head structure allows ProgNN to learn separate encoder for each task with forward connections from the past encoders to the current one (Figure 1D). This creates the possibility of having forward transfer while freezing backward transfer. Note that if the encoders are learned independently across different tasks, they may have learned useful **representations** that the tasks can mutually leverage. Thus, a further generalization of the decomposition in Figure 1B allows for each channel to ensemble the encoders (Figure 1E). Doing so requires generalizing the definition of the *channel*, so that it can operate on multiple distinct encoders. The result is that the channels *ensemble representations* (learned by the encoders), rather than decisions (learned by the channels). The channels ensemble all the existing representations, regardless of the order in which they were learned. In this scenario, like with bagging and boosting, the ensemble of channels then feeds into the single decoder. When each encoder has learned complementary representations, this latter approach has certain appealing properties, particularly in multiple task scenarios, including lifelong learning. For example, Model Zoo (Ramesh & Chaudhari, 2021) ensembles multiple encoders learned over different subsets of tasks using the boosting approach. On the other hand, we developed two different representation ensembling algorithms based on bagging of models trained on individual task. As we will show empirically, these two ensemble methods tend to outperform the existing state-of-the-art algorithms. It is shown in Wyner et al. (2017) that both bagging and boosting asymptotically converges to the Bayes optimal solution. However, for finite sample size and similar model complexity, we empirically find bagging approach to lifelong learning performs better than that of boosting when the training sample size is low (see Figure 3) whereas boosting performs better on large training sample size (See Appendix Figure 4, 7 and 8).

The key to both of our algorithms is the realization that both forests and networks partition feature space into a union of polytopes (Priebe et al., 2020). Thus, the internal representation learned by each can be considered a sparse vector encoding which polytope a given sample resides in.

In either of the algorithms, as new data from a new task arrives, our algorithm first builds a new independent encoder (using forests or networks), mapping each data point to a sparse vector encoding which polytope it is in. Then, it builds the channel for this new task, which integrates information across all existing encoders, thereby enabling forward transfer. If new data arrive from an old task, it can leverage the new encoders to update the channels from the old tasks, thereby enabling backward transfer. In either case, new test data are passed through all existing encoders and corresponding channels to make a prediction. Note that while updating the previous task channels with the cross-task posteriors, we do not need to subsample the previous task data (see Appendix D for implementation details and pseudocodes).

### 4.1 Synergistic Forests

Synergistic Forests (SynF) ensembles decision trees or forests. For each task, the encoder $u_t$ of a SynF is the representation learned by a decision forest (Amit & Geman, 1997; Breiman, 2001). The leaf nodes of each decision tree partition the input space $\mathcal{X}$ (Breiman et al., 1984). The representation of $x \in \mathcal{X}$ corresponding to a single tree can be a one-hot encoded $L_b$-dimensional vector with a 1 in the location corresponding to the leaf $x$ falls into of tree $b$. Conceptually, this corresponds to partitioning the input space into $L_b$ polytopes with $x$ falling into one of them. The representation of $x$ resulting from the collection of trees simply concatenates the $B$ one-hot vectors from the $B$ trees. Thus, the encoder $u_t$ is the mapping from $\mathcal{X}$

to a $B$-sparse vector of length $\sum_{b=1}^{B} L_b$. In other word, $u_t$ is a collection of polytopes learned over the input space. The channel then learns the class-conditional posteriors by populating the polytopes with out-of-bag samples and taking class votes, as in "honest trees" (Breiman et al., 1984; Denil et al., 2014; Athey et al., 2019). Each channel outputs the average normalized class votes across the collection of trees, adjusted for finite sample bias (Mehta et al., 2019). The decoder $w_t$ averages the posterior estimates and outputs the argmax to produce a single prediction. Recall that honest decision forests are universally consistent classifiers and regressors (Athey et al., 2019), meaning that with sufficiently large sample sizes, under suitable though general assumptions, they will converge to minimum risk. Thus, the single task version of this approaches simplifies to an approach called "Uncertainty Forests" (Mehta et al., 2019). Table 2 in the appendix lists the hyperparameters used in the CIFAR experiments.

### 4.2 Synergistic Networks

A Synergistic Network (SynN) ensembles deep networks. For each task, the encoder $u_t$ in an SynN is the "backbone" of a deep network (DN), including all but the final layer. Thus, each $u_t$ maps an element of $\mathcal{X}$ to an element of $\mathbb{R}^d$, where $d$ is the number of neurons in the penultimate layer of the DN. The channels are learned via $k$-Nearest Neighbors ($k$-NN) (Stone, 1977) over the $d$ dimensional representations of $\mathcal{X}$. The decoder is the same as above.

SynN was motivated by ProgNN, but differs from ProgNN in two key ways. First, recall that ProgNN builds a new neural network "column" for each new task, **and also builds lateral connections between the new column and all previous columns**. In contrast, SynN excludes those lateral connections, thereby greatly reducing the number of parameters and train time. Moreover, this makes each representation independent, thereby potentially avoiding interference across representations. Second, for inference on task $j$ data, assuming we have observed tasks up to $J > j$, ProgNN only leverages representations learned from tasks up to $j$, thereby excluding tasks $j + 1, \ldots, J$. In contrast, SynN leverages representations from all $J$ tasks. This difference enables backward transfer. SynF adds yet another difference as compared to SynN by replacing the deep network encoders with random forest encoders. This has the effect of making the capacity, space complexity, and time complexity scale with the complexity and sample size of each task. In contrast, both ProgNN and SynN have a fixed capacity for each task, even if the tasks have very different sample sizes and complexities.

## 5 Results

### 5.1 A computational taxonomy of lifelong learning

Lifelong learning approaches can be divided into those with fixed computational space resources, and those with growing space resources. We, therefore, quantify the computational space and time complexity of the internal representation of a number of algorithms. We also study the representation capacity of these algorithms. We use the soft-O notation $\tilde{\mathcal{O}}$ to quantify complexity (van Rooij et al., 2019). Letting $n$ be the sample size and $T$ be the number of tasks, we write that a lifelong learning algorithm is $f(n, t) = \tilde{\mathcal{O}}(g(n, T))$ when $|f|$ is bounded above asymptotically by a function $g$ of $n$ and $T$ up to a constant factor and polylogarithmic terms. Again, while calculating the space complexity, we have ignored the space required for a growing new head for the new task. Table 1 summarizes the capacity, space and time complexity of several reference algorithms, as well as our SynN and SynF. For the deep learning methods, we assume that the number of iterations is proportional to the number of samples. For space and time complexity, the table shows results as a function of $n$ and $T$, as well as the common scenario where sample size per task is fixed and therefore proportional to the number of tasks, $n \propto T$.

Parametric lifelong learning methods have a representational capacity is invariant to sample size and task number. Although the space complexity of some of these algorithms grow (because the size of the constraints grows, or they continue to store more and more data), their capacity is fixed. Thus, given a sufficiently large number of tasks, without placing constraints on the relationship between the tasks, eventually all parametric methods will catastrophically forget at least some things. EWC, Online EWC, SI, and LwF are all examples of parametric lifelong learning algorithms.

Table 1: Capacity, space, and time constraints of the representation learned by various lifelong learning algorithms. We show soft-O notation ($\tilde{\mathcal{O}}(\cdot, \cdot)$ defined in main text) as a function of $n = \sum_t^T n_t$ and $T$, as well as the common setting where $n$ is proportional to $T$. Our algorithms and DF-CNN are the only algorithms whose space and time both grow quasilinearly with capacity growing.

| Parametric | Capacity | Space | | Time | | Examples |
|---|---|---|---|---|---|---|
| | $(n, T)$ | $(n, T)$ | $(n \propto T)$ | $(n, T)$ | $(n \propto T)$ | |
| parametric | 1 | 1 | 1 | $n$ | $n$ | O-EWC, SI, LwF |
| parametric | 1 | $T$ | $n$ | $nT$ | $n^2$ | EWC |
| parametric | 1 | $n$ | $n$ | $nT$ | $n^2$ | Total Replay |
| semi-parametric | $T$ | $T^2$ | $n^2$ | $nT$ | $n^2$ | ProgNN |
| semi-parametric | $T$ | $T$ | $n$ | $n$ | $n$ | DF-CNN |
| semi-parametric | $T$ | $T+n$ | $n$ | $n$ | $n$ | SynN, Model Zoo |
| non-parametric | $n$ | $n$ | $n$ | $n$ | $n$ | SynF |

Semi-parametric algorithms' representational capacity grows slower than sample size. For example, if $T$ is increasing slower than $n$ (e.g., $T \propto \log n$), then algorithms whose capacity is proportional to $T$ are semi-parametric. ProgNN is semi-parametric, nonetheless, its space complexity $\tilde{\mathcal{O}}(T^2)$ due to the lateral connections. Moreover, the time complexity for ProgNN also scales quadratically with $n$ when $n \propto T$. Thus, an algorithm that literally stores all the data it has ever seen, and retrains a fixed size network on all those data with the arrival of each new task, would have smaller space complexity and the same time complexity as ProgNN. For comparison, we implement such an algorithm and refer to it as Total Replay. DF-CNN improves upon ProgNN by introducing a "knowledge base" with lateral connections to each new column, thereby avoiding all pairwise connections. Because these semi-parametric methods have a fixed representational capacity per task, they will either lack the representation capacity to perform well given sufficiently complex tasks, and/or will waste resources for very simple tasks. SynN eliminates the lateral connections between columns of the network, thereby reducing space complexity down to $\tilde{\mathcal{O}}(T)$. SynN stores all the data to enable backward transfer, but retains linear time complexity.

SynF is the only non-parametric lifelong learning algorithm to our knowledge. Its capacity, space and time complexity are all $\tilde{\mathcal{O}}(n)$, meaning that its representational capacity naturally increases with the complexity of each task.

## 5.2 Illustrating Synergistic Learning with SynF and SynN

In this paper, we have proposed two approaches of synergistic lifelong learning algorithm, namely- SynF and SynN. They are based on the same representation ensembling method illustrated in Fig. 1. For SynN, we have used the architecture described in van de Ven et al. (2020) as "5 convolutional layers followed by 2 fully-connected layers each containing 2,000 nodes with ReLU non-linearities and a softmax output layer" as encoder. We trained this network using cross-entropy loss and the Adam optimizer (Kingma & Ba, 2014) to learn the encoder. The channels are learned via $k$-Nearest Neighbors ($k$-NN). Recall that a $k$-NN, with $k$ chosen such that as the number of samples goes to infinity, $k$ also goes to infinity, while $\frac{k}{n} \to 0$, is a universally consistent classifier (Stone, 1977). We use $k = 16 \log_2 n$, which satisfies these conditions. However, SynN requires training DNs as encoders which is computationally expensive to do it for many Monte Carlo repetitions. Therefore, for SynN experiments we did 100 repetitions and reported the results after smoothing it using moving average with a window size of 5. Again, for the SynF experiments we used 1000 repetitions and reported the mean of these repetitions.

### 5.2.1 Synergistic learning in a simple environment

Consider a very simple two-task environment: Gaussian XOR and Gaussian Exclusive NOR (XNOR) (Figure 2A, see Appendix F for details). The two tasks share the exact same discriminant boundaries: the coordinate

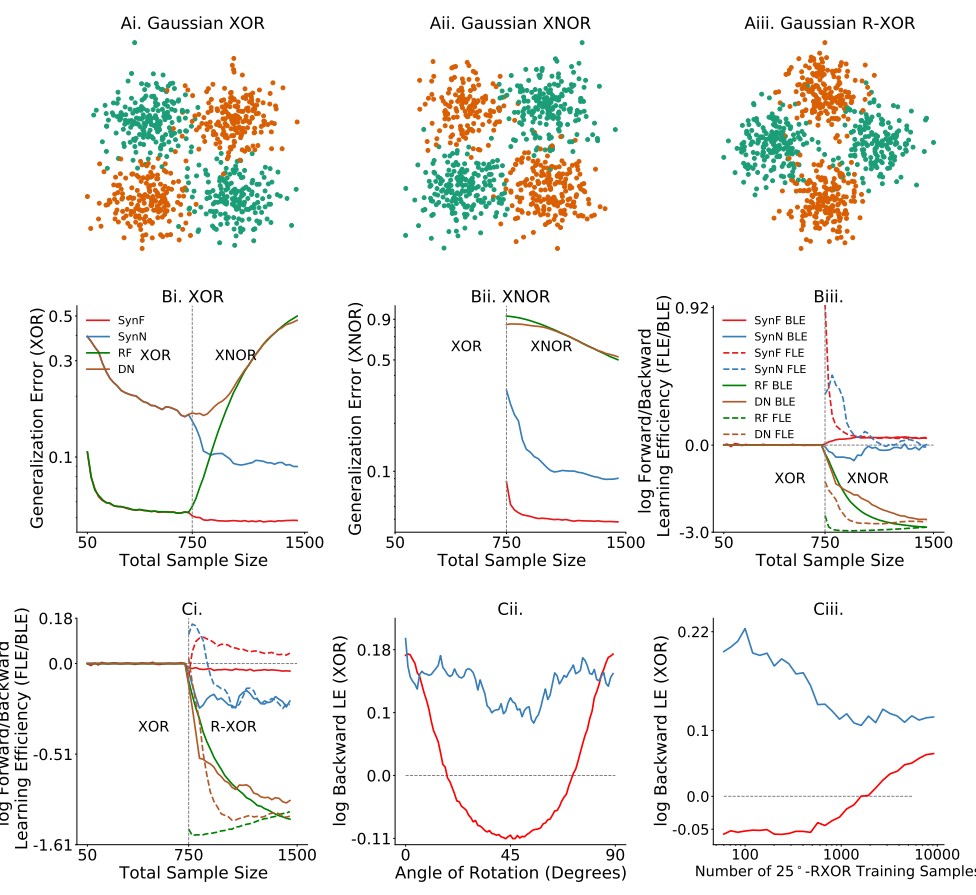

Figure 2: **Synergistic Forest and Synergistic Network demonstrate forward and backward transfer.** (*A*) 750 samples from: (*Ai*) Gaussian XOR, (*Aii*) XNOR, which has the same optimal discriminant boundary as XOR, and (*Aiii*) R-XOR, which has a discriminant boundary that is uninformative, and therefore adversarial, to XOR. (*Bi*) Generalization error for XOR, and (*Bii*) XNOR of both SynF (red), RF (green), SynN(blue), DN (dark orange). SynF outperforms RF on XOR when XNOR data is available, and on XNOR when XOR data are available. The same result is true for SynN sand DN. (*Biii*) Forward and backward learning efficiency of SynF are positive for all sample sizes, and are negative for all sample sizes for RF. Again, FLE and BLE is higher for SynNcompared to those of DN. (*Ci*) In an adversarial task setting (XOR followed by R-XOR), SynFand SynN gracefully forgets XOR, whereas RFand DN demonstrate catastrophic forgetting and interference. (*Cii*) log BLE with respect to XOR is positive when the optimal decision boundary of $\theta$-XOR is similar to that of XOR (e.g. angles near 0° and 90°), and negative when the discriminant boundary is uninformative, and therefore adversarial, to XOR (e.g. angles near 45°) . (*Ciii*) BLE is a nonlinear function of the source training sample size (right).

axes. Thus, transferring from one task to the other merely requires learning a bit flip. We sample a total 750 samples from XOR, followed by another 750 samples from XNOR.

SynF and random forests (RF) achieve the same generalization error on XOR when training with XOR data (Figure 2Bi). But because RF does not account for a change in task, when XNOR data appear, RF performance on XOR deteriorates (it catastrophically forgets). In contrast, SynF continues to improve on XOR given XNOR data, demonstrating backward transfer. Now consider the generalization error on *XNOR* (Figure 2Bii). Both SynF and RF are at chance levels for XNOR when only XOR data are available. When XNOR data are available, RF must unlearn everything it learned from the XOR data, and thus its performance on XNOR starts out nearly maximally inaccurate, and quickly improves. On the other hand,

because SynF can leverage the encoder learned using the XOR data, upon getting *any* XNOR data, it immediately performs quite well, and then continues to improve with further XNOR data, demonstrating forward transfer (Figure 2Biii). SynF demonstrates positive forward and backward transfer for all sample sizes, whereas RF fails to demonstrate forward or backward transfer, and eventually catastrophically forgets the previous tasks. Similar results are visible for SynN and DN in Figure 2.

### 5.2.2 Synergistic learning in adversarial environments

Statistics has a rich history of *robust learning* (Huber, 1996; Ramoni & Sebastiani, 2001), and machine learning has recently focused on *adversarial learning* (Szegedy et al., 2014; Zhang et al., 2018; 2020; Lowd & Meek, 2005). However, in both cases the focus is on adversarial *examples*, rather than adversarial *tasks*. In the context of synergistic learning, we informally define a task $t$ to be adversarial with respect to task $t'$ if the true joint distribution of task $t$, without any domain adaptation, impedes performance on task $t'$. In other words, training data from task $t$ can only add noise, rather than signal, for task $t'$. An adversarial task for Gaussian XOR is Gaussian XOR rotated by 45° (R-XOR) (Figure 2Aiii). Training on R-XOR therefore impedes the performance of SynF and SynN on XOR, and thus backward transfer falls below one, demonstrating graceful forgetting (Aljundi et al., 2018) (Figure 2Ci). Because R-XOR is more difficult than XOR for SynF (because the discriminant boundaries are oblique (Tomita et al., 2020)), and because the discriminant boundaries are learned imperfectly with finite data, data from XOR can actually improve performance on R-XOR, and thus forward transfer is positive. In contrast, both forward and backward transfer are negative for RF and DN.

To further investigate this relationship, we design a suite of R-XOR examples, generalizing R-XOR from only 45° to any rotation angle between 0° and 90°, sampling 100 points from XOR, and another 100 from each R-XOR (Figure 2Cii). As the angle increases from 0° to 45°, log BLE flips from positive ($\approx 0.18$) to negative ($\approx -0.11$) for SynF. A similar trend is also visible for SynN. The 45°-XOR is the maximally adversarial R-XOR. Thus, as the angle further increases, log BLE increases back up to $\approx 0.18$ at 90°, which has an identical discriminant boundary to XOR. Moreover, when $\theta$ is fixed at 25°, BLE increases at different rates for different sample sizes of the source task (Figure 2Ciii).

Together, these experiments indicate that the amount of transfer can be a complicated function of (i) the difficulty of learning good representations for each task, (ii) the relationship between the two tasks, and (iii) the sample size of each. Appendix F further investigates this phenomenon in a multi-spiral environment.

### 5.3 Real data experiments

We consider two modalities for real data experiments: vision and language. Below we provide a detailed analysis of the performance of lifelong learning algorithms in vision data; Appendix G provides details for our language experiments, which have qualitatively similar results illustrating that SynF and SynN are modality agnostic, sample and computationally efficient, lifelong learning algorithms. We also provide additional vision experiments on larger datasets in the appendix which shows the relative performance gain of Model Zoo (boosting) compared to that of our approach (bagging) on large datasets. However, under the lifelong learning framework, a learning agent, constrained by capacity and computational time, is sequentially trained on multiple tasks. For each task, it has access to limited training samples (Chen & Liu, 2016; Lee et al., 2019), and it improves on a particular task by leveraging knowledge from the other tasks. Therefore, for our following experiments, we are particularly interested in the behavior of our representation ensembling algorithms in the low training sample size regime. The below experiments use only 500 training samples per task. For the corresponding experiments using higher training samples per task (5,000 samples), see Appendix Figure 4.

The CIFAR 100 challenge (Krizhevsky, 2012), consists of 50,000 training and 10,000 test samples, each a 32x32 RGB image of a common object, from one of 100 possible classes, such as apples and bicycles. CIFAR 10x10 divides these data into 10 tasks, each with 10 classes (Lee et al., 2019) (see Appendix G for details). We compare SynF and SynN to the deep lifelong learning algorithms discussed above. In the subsequent experiments, we have reported the average accuracy over all the tasks as more tasks are seen as proposed in (Lomonaco & Maltoni, 2017; Maltoni & Lomonaco, 2019) for both the lifelong and single task learners along

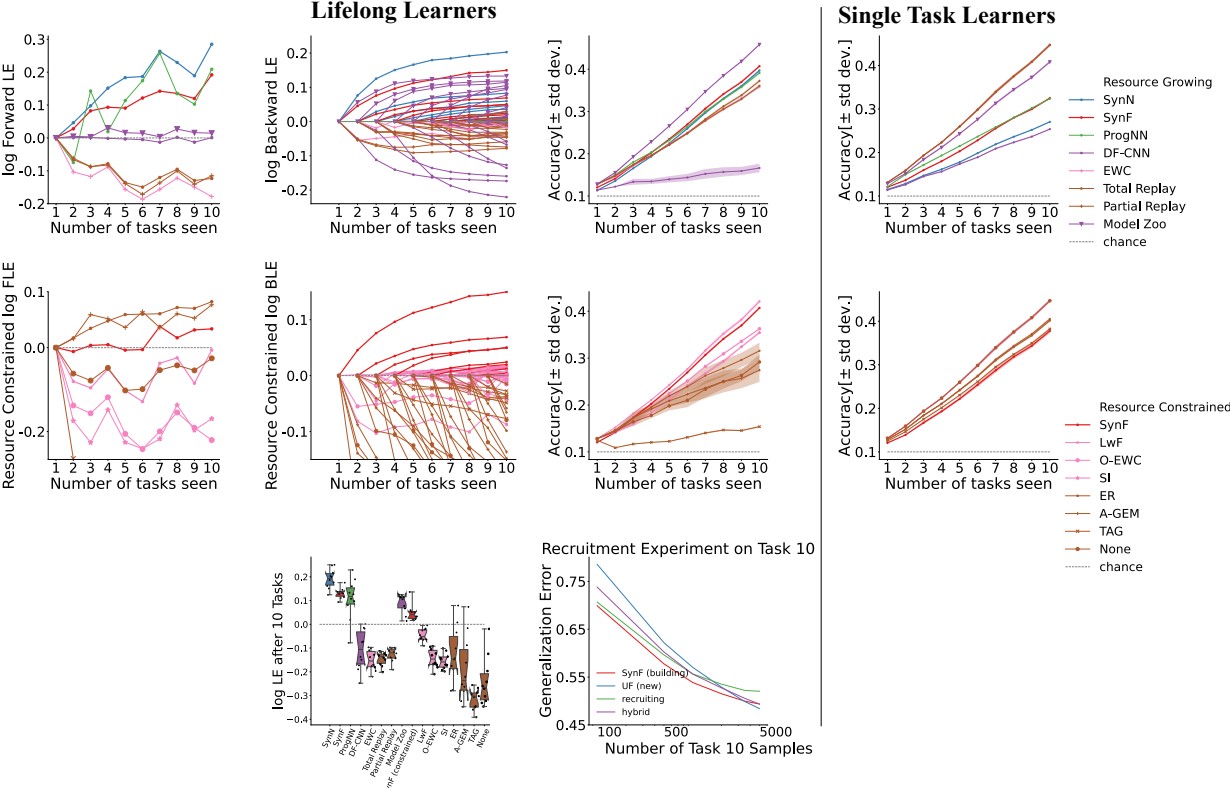

Figure 3: **Performance of different algorithms on the CIFAR 10x10 vision experiments.** *Top row*: Forward and backward transfer efficiency for various resource building algorithms. SynF and SynN consistently demonstrate both forward and backward transfer for each task, whereas ProgNN and DF-CNN do not. Rightmost two columns show the average global accuracy after each task for both the lifelong and the single task learner. In all of the plots, the performance of the chance algorithm which chooses a label at random is shown as a horizontal dashed line along 0. *Middle row*: Same as above but comparing each algorithm with a fixed amount of resources. SynF is the only approach that demonstrate forward or backward transfer. *Bottom left*: Transfer efficiencies of various algorithms for the 10 tasks after seeing the 10-th task. Both SynN and SynF synergistically learn over all the 10 tasks whereas other algorithms (except ProgNN) catastrophically forget. *Bottom right*: Building and recruiting ensembles are two boundaries of a continuum, with hybrid models in the middle. SynF achieves lower (better) generalization error than other approaches until 5,000 training samples on the new task are available, but eventually a hybrid approach wins.

with our proposed learning efficiencies. However, only multitask accuracy cannot ascertain the superiority of an algorithm. For example, note that in Figure 3 middle row third column, `LwF` has better average global accuracy compared to that of `SynF`. However, as shown in the rightmost column of the middle row of Figure 3, `LwF` has higher single task accuracy, i.e., accuracy when the learner has access to a single task data only. Therefore, `LwF` improves accuracy for each task without doing meaningful transfer of information between the tasks. This is evident from the forward and backward learning efficiency curves in middle row of Figure 3. For the `FLE` curves, we report forward learning efficiency on the corresponding task as that task is introduced. Again for the backward learning efficiency, we evaluate the backward learning efficiency on all of the tasks introduced so far as a new task is introduced. Therefore, for each task the log(`BLE`) curve starts from 0 when the corresponding task is introduced and goes upward (positive) or downward (negative) as more tasks are seen.

### 5.3.1 Resource Growing Experiments

We first compare `SynF` and `SynN` to state-of-the-art resource growing algorithms: Model Zoo, ProgNN and DF-CNN (Figure 3, top panels). Both `SynF` and `SynN` demonstrate positive forward transfer for every task (`SynF` increases nearly monotonically), indicating they are robust to distributional shift in ways that ProgNN and DF-CNN are not. `SynN`, `SynF` and Model Zoo demonstrate positive backward transfer, `SynN` is actually monotonically increasing, indicating that with each new task, performance on all prior tasks increases (and `SynF` nearly monotonically increases BLE as well). In contrast, while neither ProgNN nor DF-CNN exhibit catastrophic forgetting, they also do not exhibit any positive backward transfer. Final transfer efficiency per task is the transfer efficiency associated with that task having seen all the data. `SynF` and `SynN` both demonstrate positive final transfer efficiency for all tasks (synergistic learning), whereas ProgNN and DF-CNN both exhibit negative final transfer efficiency for at least one task.

### 5.3.2 Resource Constrained Experiments

It is possible that the above algorithms are leveraging additional resources to improve performance without meaningfully transferring information between representations. To address this concern, we devised a "resource constrained" variant of `SynF`. In this constrained variant, we compare the lifelong learning algorithm to its single task variant, but ensure that they both have the same amount of resources. For example, on Task 2, we would compare `SynF` with 20 trees (10 trained on 500 samples from Task 1, and another 10 trained on 500 samples from Task 2) to `RF` with 20 trees (all trained on 500 samples Task 2). If `SynF` is able to meaningfully transfer information across tasks, then its resource-constrained FLE and BLE will still be positive. Indeed, FLE remains positive after enough tasks, and BLE is actually invariant to this change (Figure 3, bottom left and center). In contrast, all of the reference algorithms that have fixed resources exhibit negative forward and backward transfer. Moreover, the reference algorithms also all exhibit negative final transfer efficiency on each task, whereas our resource constrained `SynF` maintains positive final transfer on every task (Figure 3, top right). Interestingly, when using 5,000 samples per task, total and partial replay methods are able to demonstrate positive forward and backward transfer (Supplementary Figure 4), although they require quadratic time. Note that in this experiment, building the single task learners actually requires substantially *more* resources, specifically, $10 + 20 + \cdots + 100 = 550$ trees, as compared with only 100 trees in the prior experiments. In general, to ensure single task learners use the same amount of resources per task as omnidirectional learners requires $\tilde{\mathcal{O}}(n^2)$ resources, where as `SynF` only requires $\tilde{\mathcal{O}}(n)$, a polynomial reduction in resources.

In both cases, resource growing or resource constrained, both `SynF` and `SynN` show synergistic learning over all the 10 tasks (Figure 3, top right panel) whereas all other algorithms except ProgNN suffer from catastrophic forgetting.

### 5.3.3 Resource Recruiting Experiments

The binary distinction we made above, algorithms either build resources or reallocate them, is a false dichotomy, and biologically unnatural. In biological learning, systems develop from building (juvenile) to constrained (adult) resources (which requires recruiting some resources for new tasks). We therefore train

SynF on the first nine CIFAR 10x10 tasks using 50 trees per task, with 500 samples per task. For the tenth task, we could (i) select the 50 trees (out of the 450 existing trees) that perform best on task 10 (recruiting), (ii) train 50 new trees, as SynF would normally do (building), (iii) build 25 and recruit 25 trees (hybrid), or (iv) ignore all prior trees (RF). SynF outperforms other approaches except when 5,000 training samples are available, but the recruiting approach is nearly as good as SynF (Figure 3, bottom right). This result motivates future work to investigate optimal strategies for determining how to optimally leverage existing resources given a new task, and task-unaware settings.

### 5.3.4 Adversarial Experiments

Consider the same CIFAR 10x10 experiment above, but, for tasks two through nine, randomly permute the class labels within each task, rendering each of those tasks adversarial with regard to the first task (because the labels are uninformative). Figure 4A indicates that BLE for both SynF and SynN is invariant to such label shuffling (the other algorithms also seem invariant to label shuffling, but did not demonstrate positive backward transfer). Now, consider a Rotated CIFAR experiment, which uses only data from the first task, divided into two equally sized subsets (making two tasks), where the second subset is rotated by different amounts (Figure 4, right). Learning efficiency of both SynF and SynN is nearly invariant to rotation angle, whereas the other approaches are far more sensitive to rotation angle. Note that zero rotation angle corresponds to the two tasks *having identical distributions*.

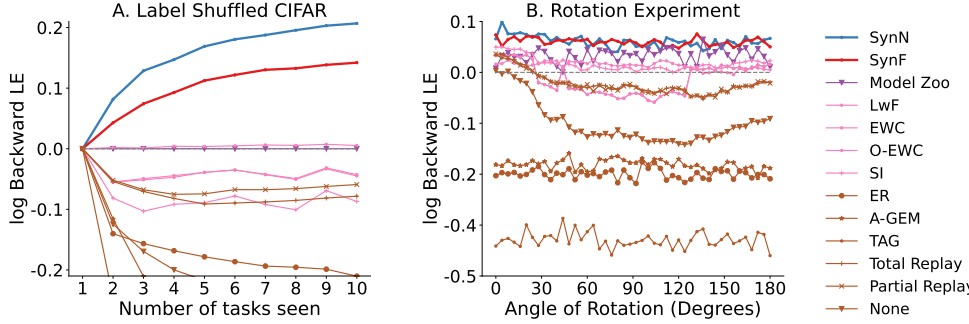

Figure 4: **Extended CIFAR 10x10 experiments.** (*A*) Shuffling class labels within tasks two through nine with 500 samples each demonstrates both SynF and SynN can still achieve positive backward transfer, and that the other algorithms still fail to transfer. (*B*) SynF and SynN are nearly invariant to rotations, whereas other approaches are more sensitive to rotation.

## 6 Discussion

We introduced quasilinear representation ensembling as an approach to synergistic lifelong learning. Two specific algorithms, SynF and SynN, achieve both forward and backward transfer, due to leveraging resources (encoders) learned for other tasks without undue computational burdens. Forest-based representation ensembling approaches can easily add new resources when appropriate. This work therefore motivates additional work on deep learning to enable dynamically adding resources when appropriate (Yoon et al., 2017).

To achieve backward transfer, SynF and SynN stored old data to vote on the newly learned transformers. Because the representation space scales quasilinearly with sample size, storing the data does not increase the space complexity of the algorithm, and it remains quasilinear. It could be argued that by keeping old data and training a model with increasing capacity from scratch (a sequential multitask learning approach), it would be straightforward to maintain performance (TE = 1) in a particular task. However, it is not obvious how to achieve backward transfer with quasilinear time and space complexity even if we are allowed to store all the past data, because computational time would naively become quadratic. For example, both ProgNN and Total Replay have quadratic time complexity, unlike SynF and SynN. Thus, one natural extension of this work would obviate the need to store all the data by using a generative model.

While we employed quasilinear representation ensembling to address catastrophic forgetting, the paradigm of ensembling *representations* rather than *learners* can be readily applied more generally. For example, "batch effects" (sources of variability unrelated to the scientific question of interest) have plagued many fields of inquiry, including neuroscience (Bridgeford et al., 2020) and genomics (Johnson et al., 2007). Similarly, federated learning is becoming increasingly central in artificial intelligence, due to its importance in differential privacy (Dwork, 2008). This may be particularly important in light of global pandemics such as COVID-19, where combining small datasets across hospital systems could enable more rapid discoveries (Vogelstein et al., 2020).

Finally, our quasilinear representation ensembling approach closely resembles the constructivist view of brain development (Quartz, 1999; Karmiloff-Smith, 2017). According to this view, the brain goes through progressive elaboration of neural circuits resulting in an augmented cognitive representation while maturing in a certain skill. In a similar way, representation ensembling algorithms can mature in a particular skill such as vision tasks by learning a rich encoder dictionary from different vision datasets and thereby, transfer forward to future or yet unseen vision dataset (see CIFAR 10x10 recruitment experiment as a proof). However, there is also substantial pruning during development and maturity in the brain circuitry which is important for performance (Sakai, 2020). This motivates future work for pruning adversarial encoders to enhance the transferability among tasks even more. Moreover, by carefully designing experiments in which both behaviors and brain are observed while learning across sequences of tasks (possibly in multiple stages of neural development or degeneration), we may be able to learn more about how biological agents are able to synergistically learn so efficiently, and transfer that understanding to building more effective artificial intelligences. In the meantime, our code, including code to reproduce the experiments in this manuscript, is available from `http://proglearn.neurodata.io/`.

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

## A  A Concrete Example on Lifelong Learning Metrics

In this section, we propose that it is desirable for lifelong learning metrics to have a few properties that are not currently present in the existing metrics. First, the amount of overall transfer should be decomposable into forward and backward components. This enables one to report an overall level of transfer if desired as a summary, rather than requiring yet a third metric to quantify the amount of overall transfer. Please see figure 3 last row first panel where we report final learning efficiency after 10 tasks as a summary of the overall performance of the model.

Second, one should be able to generally define transfer learning, and then obtain forward and backward as two specific interesting special cases. This is because both forward and backward transfer are special cases

of transfer learning achieved from two different streams of data, i.e., past task data and future task data, respectively. In our proposed metrics, both BLE and FLE are defined by the same exact function, just with different data streams in the numerator and the denominator.

Third, the amount of transfer should be dependent on the accuracy level of the algorithms. This is because in general, once we get to high accuracy levels (e.g., 98% or so), we care deeply about gains in *relative* performance, that is, reducing error from 2% to 1% is a big deal. In contrast, if one reduces error from 49% to 48%, that is relatively less interesting and impactful. Recall that for Bernoulli random variables, the variance of the estimator is a function of how close it is to 50%. In both of the cases, the change in accuracy is 1% whereas learning efficiencies (LE) are 2 and 1.02, respectively. Therefore, our proposed metrics automatically account for the relative difficulties of transfer learning at different accuracy levels.

Fourth, the metrics should be able to resolve the overall transfer learning into individual task. For example, while reporting an average overall transfer a task with extremely high transfer may mask the poor performance over the other tasks.

Fifth, the metrics should validly quantify the amount of actual *transfer* of information from one (set of) data to another, rather than merely improvement. The overall accuracy over the tasks can improve simply because subsequent tasks are easier, for example, which does not indicate whether there has been any transfer. However, as we will show in section 5.3, only accuracy or differences in accuracy cannot guarantee the agent has achieved any amount of transfer. The metrics that we propose here satisfy all five of the above desiderata; we show below that existing metrics do not satisfy some of them.

In what follows we compare and contrast our proposed metrics with the metrics, namely, accuracy, backward transfer (BWT) and forward transfer (FWT), as proposed in Díaz-Rodríguez et al. (2018) on a hypothetical scenario. Consider a lifelong learning environment with two tasks each having two classes. The tasks are introduced sequentially with $n_1$ samples from Task 1 and then $n_2$ samples from Task 2. The agent has a generalization error of $R^1(f(S_n^1)) = 0.3$ on Task 1 while it has access to the Task 1 dataset only, and a generalization error of $R^2(f(S_n^2)) = 0.4$ on Task 2, while it has access to the Task 2 dataset only. Now consider the scenario when the agent has the same hyper-parameters and sequential access to all the task datasets. Suppose the model has the generalization error on two tasks enumerated as in table 1. Note that the FLEs are given by: $\mathsf{FLE}_n^1(f) = \frac{R^1(f(\mathbf{S}_n^1))}{R^1(f(\mathbf{S}_n^{\leq 1}))} = 1$ and $\mathsf{FLE}_n^2(f) = \frac{R^2(f(\mathbf{S}_n^2))}{R^2(f(\mathbf{S}_n^{\leq 2}))} = 0.89$. The performance metrics can be summarized as in table 1.

Table 1: Learning metrics summarized on a hypothetical scenario.

| Number of samples seen $n$ | n = $n_1$ | $n = n_1 + n_2$ |
|---|---|---|
| $R^1(\mathbf{S}_n)$ | 0.30 | 0.32 |
| $R^2(\mathbf{S}_n)$ | 0.5 | 0.45 |
| $\mathsf{BLE}_n^1(f)$ | 1 | 0.94 |
| $\mathsf{FLE}_n^1(f)$ | 1 | 1 |
| $\mathsf{LE}_n^1(f) = \mathsf{BLE}_n^1(f) \times \mathsf{FLE}_n^1(f)$ | 1 | 0.94 |
| $\mathsf{BLE}_n^2(f)$ | − | 1 |
| $\mathsf{FLE}_n^2(f)$ | − | 0.89 |
| $\mathsf{LE}_n^2(f) = \mathsf{BLE}_n^2(f) \times \mathsf{FLE}_n^2(f)$ | − | 0.89 |
| Average global accuracy | 0.60 | 0.62 |
| BWT | 0 | −0.02 |
| FWT | 0 | 0.5 |

As evident in Table 1, the transfer learning for Task 1 comes from backward transfer from Task 2 whereas for Task 2 it comes from forward learning from Task 1. As a summary, one can look at the final LEs over all the tasks after all tasks have been introduced. Note that in Table 1 the learning efficiencies are never greater than 1. However, the global accuracy increased from 60% to 62%. Therefore, only using multi-task accuracy may falsely detect positive transfer. In Table 1, BWT can correctly identify an overall negative backward

transfer or forgetting. However, being an average quantity, it can not resolve the overall backward transfer into individual task. As a result, a task with extremely high backward transfer may mask all the negative backward transfer from the other tasks giving a net positive transfer. Also, note that FWT considers the zero-shot accuracies for the tasks. However, for supervised learning, the model will randomly assign labels for a task if it does not have any information about the labels in the corresponding task. In other words, in a multi-headed lifelong learning model, growing a task specific head is necessary to evaluate the model on a specific task. Otherwise, the zero-shot accuracy will always be at chance which is also shown in the Table 1 of Díaz-Rodríguez et al. (2018).

## B  Decision Tree as a Compositional Hypothesis

Consider learning a decision tree for a two class classification problem. The input to the decision tree is a set of $n$ feature-vector/response pairs, $(x_i, y_i)$. The learned tree structure corresponds to the encoder $u$, because the tree structure maps each input feature vector into an indicator encoding in which leaf node each feature vector resides. Formally, $u : \mathcal{X} \mapsto [L]$, where $[L] = \{1, 2, \ldots, L\}$ and $L$ is the total number of leaf nodes. In other words, $u$ maps from the original data space, to a $L$-dimensional one-hot encoded sparse binary vector, where the sole non-zero entry indicates in which leaf node a particular observation falls, that is, $\tilde{x} := u(x) \in \{0, 1\}^L$ where $\|\tilde{x}\| = 1$.

Learning the voter is simply a matter of counting the fraction of observations in each leaf per class. So, the voter is trained using $n$ pairs of transformed feature-vector/response pairs $(\tilde{x}_i, y_i)$, and it assigns a probability of each class in each leaf: $\{v_l := \mathbb{P}[y_i = 1 | \tilde{x}_i = l], \forall l \in [L]\}$ and $v(\tilde{x}) = v_{\tilde{x}}$. In other words, for two class classification, $v$ maps from the $L$-dimensional binary vector to the probability that $x$ is in class 1. The decider is simply $w(v(\tilde{x})) = \mathbb{1}_{\{v(\tilde{x}) > 0.5\}}$, that is, it outputs the most likely class label of the leaf node that $x$ falls into.

For inference, the tree is given a single $x$, and it is passed down the tree until it reaches a leaf node, where it is represented by its leaf identifier $\tilde{x}$. The voter takes $\tilde{x}$ as input, and outputs the estimated posterior probability of being in class 1 for the leaf node in which $\tilde{x}$ resides: $v(\tilde{x}) = \mathbb{P}[y = 1 | \tilde{x}]$. If $v(\tilde{x})$ is bigger than 0.5, the decider decides that $x$ is in class 1, and otherwise, it decides it is in class 0.

## C  Compositional Representation Ensembling

Consider a scenario in which we have two tasks, one following the other. Assume that we already learned a single decomposable hypothesis for the first task: $w_1 \circ v_1 \circ u_1$, and then we get new data associated with a second task. Let $n_1$ denote the sample size for the first task, and $n_2$ denote the sample size for the second task, and $n = n_1 + n_2$. The representation ensembling approach generally works as follows. First, since we want to transfer forward to the second task, we push all the new data through the first encoder $u_1$, which yields $\tilde{x}_{n_1+1}^{(1)}, \ldots, \tilde{x}_n^{(1)}$. Second, we learn a new encoder $u_2$ using the new data, $\{(x_i, y_i)\}_{i=n_1+1}^n$. We then push the new data through the new encoder, yielding $\tilde{x}_{n_1+1}^{(2)}, \ldots, \tilde{x}_n^{(2)}$. Third, we train a new channel, $v_2$. To do so, $v_2$ is trained on the outputs from both encoders, that is, $\{(\tilde{x}_i^{(j)}, y_i)\}_{i=n_1+1}^n$ for $j = 1, 2$. The output of $v_2$ for any new input $x$ is the posterior probability (or score) for that point for each potential response in task two (class label). Thus, by virtue of ensembling these representations, this approach enables forward transfer (Rusu et al., 2016; Dhillon et al., 2020).

Now, we would also like to improve performance on the first task using the second task's data. While many lifelong methods have tried to achieve this kind of backward transfer, to date, they have mostly failed (Ruvolo & Eaton, 2013). Recall that previously we had already pushed all the first task data through the first task encoder, which had yielded $\tilde{x}_1^{(1)}, \ldots, \tilde{x}_{n_1}^{(1)}$. Assuming we kept any of the first task's data, or can adequately simulate it, we can push those data through $u_2$ to get a second representation of the first task's data: $\tilde{x}_1^{(2)}, \ldots, \tilde{x}_{n_1}^{(2)}$. Then, $v_1$ would be trained on both representations of the first task's data. This 'replay-like' procedure facilitates backward transfer, that is, improving performance on previous tasks by leveraging data from newer tasks. Both the forward and backward transfer updates can be implemented every time we

---

**Algorithm 1** Add a new SYNX encoder for a task. OOB = out-of-bag.

---

**Require:**

    (1) $t$                                         ▷ current task number

    (2) $\mathcal{D}_n^t = (\mathbf{x}^t, \mathbf{y}^t) \in \mathbb{R}^{n \times p} \times \{1, \ldots, K\}^n$           ▷ training data for task $t$

**Ensure:**

    (1) $u_t$                                       ▷ a encoder set

    (2) $\mathcal{I}_{OOB}^t$                           ▷ a set of the indices of OOB data

 1: **function** SYNX.FIT($t, (\mathbf{x}^t, \mathbf{y}^t)$)

 2:     $u_t, \mathcal{I}_{OOB}^t \leftarrow$ X.fit($\mathbf{x}^t, \mathbf{y}^t$)          ▷ train an encoder X on bootstrapped data

 3:     **return** $u_t, \mathcal{I}_{OOB}^t$

 4: **end function**

---

**Algorithm 2** Add a new SYNX channel for the current task.

---

**Require:**

    (1) $t$                                          ▷ current task number

    (2) $\boldsymbol{u}_t = \{u_t\}_{t'=1}^t$                          ▷ the set of encoders

    (3) $\mathcal{D}_n^t = (\mathbf{x}^t, \mathbf{y}^t) \in \mathbb{R}^{n \times p} \times \{1, \ldots, K\}^n$        ▷ training data for task $t$

    (4) $\mathcal{I}_{OOB}^t$                ▷ a set of the indices of OOB data for the current task

**Ensure:** $\boldsymbol{v}_t = \{v_{t,t'}\}_{t'=1}^t$      ▷ in-task ($t' = t$) and cross-task ($t' \neq t$) channels for task $t$

 1: **function** SYNX.ADD__CHANNEL($t, \boldsymbol{u}_t, (\mathbf{x}_t, \mathbf{y}_t), \mathcal{I}_{OOB}^t$)

 2:     $v_{tt} \leftarrow u_t$.add_channel($(\mathbf{x}_t, \mathbf{y}_t), \mathcal{I}_{OOB}^t$)      ▷ add the in-task channel using OOB data

 3:     **for** $t' = 1, \ldots, t-1$ **do**          ▷ update the cross task channels for task $t$

 4:         $v_{tt'} \leftarrow u_{t'}$.add_channel($\mathbf{x}_t, \mathbf{y}_t$)

 5:     **end for**

 6:     **return** $v_t$

 7: **end function**

---

obtain data associated with a new task. Enabling the channels to ensemble *omnidirectionally* between all sets of tasks is the key innovation of our proposed synergistic learning approaches.

# D   Synergistic Algorithms

We propose two concrete synergistic algorithms, Synergistic Forests (SYNF) and Synergistic Networks (SYNN). The two algorithms differ in their detais of how to update representers and voters, but abstracting a level up they are both special cases of the same procedure. Let SYNX refer to any possible synergistic algorithm. Algorithms 1, 2, 3, and 4 provide pseudocode for adding representers, updating voters, and making predictions for any SYNX algorithm; the below sections provide SYNF and SYNN specific details.

Table 2: Hyperparameters for SYNF in CIFAR experiments. n_estimators is denoted by $B$, the number of trees, above.

| Hyperparameters | Value |
|---|---|
| n_estimators (500 training samples per task) | 10 |
| n_estimators (5000 training samples per task) | 40 |
| max_depth | 30 |
| max_samples (OOB split) | 0.67 |
| min_samples_leaf | 1 |

---

**Algorithm 3** Update SynX channel for the previous tasks.

---

**Require:**
    (1) $t$                                                         $\triangleright$ current task number
    (2) $u_t$                                                $\triangleright$ encoder for the current task
    (3) $\mathcal{D} = \{\mathcal{D}^{t'}\}_{t'=1}^{t-1}$                   $\triangleright$ training data for tasks $t' = 1, \cdots, t-1$
**Ensure:** $\boldsymbol{v} = \{\boldsymbol{v}_{t'}\}_{t'=1}^{t-1}$                     $\triangleright$ all previous task voters
 1:  **function** SynX.UPDATE_CHANNEL($t, u_t, \mathcal{D}$)
 2:     **for** $t' = 1, \ldots, t-1$ **do**               $\triangleright$ update the cross task channels
 3:         $v_{t't} \leftarrow u_t.\text{get\_channel}(\mathbf{x}_{t'}, \mathbf{y}_{t'})$
 4:     **end for**
 5:     **return** $\boldsymbol{v}$
 6:  **end function**

---

---

**Algorithm 4** Predicting a class label using SynX.

---

**Require:**
    (1) $x \in \mathbb{R}^p$                                               $\triangleright$ test datum
    (2) $t$                                          $\triangleright$ task identity associated with $x$
    (3) $\boldsymbol{u}$                                        $\triangleright$ all $T$ reperesenters
    (4) $\boldsymbol{v}_t$                                      $\triangleright$ channel for task $t$
**Ensure:** $\hat{y}$                                   $\triangleright$ a predicted class label
 1:  **function** $\hat{y} = $ SynX.PREDICT($t, x, v_t$)
 2:     $T \leftarrow$ SynX.get_task_number()            $\triangleright$ get the total number of tasks
 3:     $\hat{\mathbf{p}}_t = \mathbf{0}$                       $\triangleright$ $\hat{\mathbf{p}}_t$ is a $K$-dimensional posterior vector
 4:     **for** $t' = 1, \ldots, T$ **do**        $\triangleright$ aggregate the posteriors calculated from $T$ task channels
 5:         $\hat{\mathbf{p}}_t \leftarrow \hat{\mathbf{p}}_t + v_{tt'}.\text{predict\_proba}(u_{t'}(x))$
 6:     **end for**
 7:     $\hat{\mathbf{p}}_t \leftarrow \hat{\mathbf{p}}_t / T$
 8:     $\hat{y} = \arg\max_i(\hat{\mathbf{p}}_t)$     $\triangleright$ find the index $i$ of the elements in the vector $\hat{\mathbf{p}}_t$ with maximum probability
 9:     **return** $\hat{y}$
10:  **end function**

---

# E   Reference Algorithm Implementation Details

The same network architecture was used for all compared deep learning methods. Following van de Ven et al. (2020), the 'base network architecture' consisted of five convolutional layers followed by two-fully connected layers each containing 2000 nodes with ReLU non-linearities and a softmax output layer. The convolutional layers had 16, 32, 64, 128 and 254 channels, they used batch-norm and a ReLU non-linearity, they had a 3x3 kernel, a padding of 1 and a stride of 2 (except the first layer, which had a stride of 1). This architecture was used with a multi-headed output layer (i.e., a different output layer for each task) for all algorithms using a fixed-size network. For ProgNN and DF-CNN the same architecture was used for each column introduced for each new task, and in our SynN this architecture was used for the transformers $u_t$ (see above). In these implementations, ProgNN and DF-CNN have the same architecture for each column introduced for each task. Each column has an input layer followed by 4 convolutional layer with size $3 \times 3 \times 32$, $3 \times 3 \times 32$, $3 \times 3 \times 64$ and $3 \times 3 \times 64$, respectively. It is followed by a fully-connected layer with 64 nodes and an output layer with 10 nodes. ReLU activation was used after each layer. The other algorithms use a common architecture with input layers defined by the size of the input data, two hidden layers with 400 nodes each and a multi-headed output layer (different output layers for different tasks). Different algorithms only differ in the way they penalize the update of network parameters for the current task based on the previous tasks. Each of these algorithms has 1.4M parameters in total.

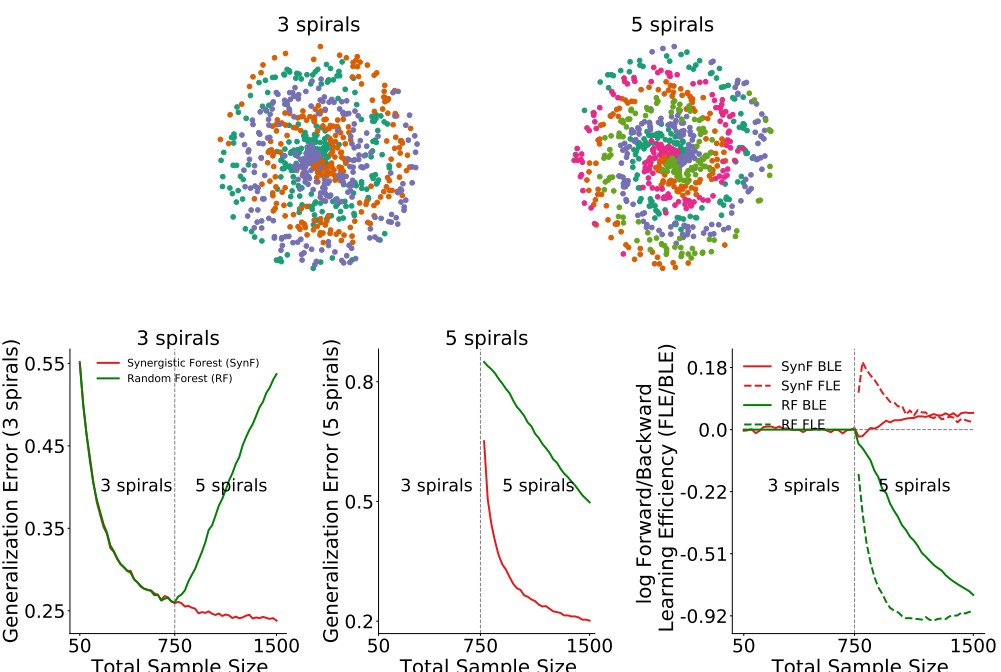

Figure 1: *Top*: 750 samples from 3 spirals (left) and 5 spirals (right). *Bottom left*: SynF outperforms RF on 3 spirals when 5 spirals data is available, demonstrating *backward* transfer in SynF. *Bottom center*: SynF outperforms RF on 5 spirals when 3 spirals data is available, demonstrating *forward* transfer in SynF. *Bottom right*: Transfer Efficiency of SynF. The forward (solid) and backward (dashed) curves are the ratio of the generalization error of SynF to RF in their respective figures. SynF demonstrates decreasing forward transfer and increasing backward transfer in this environment.

## F   Simulated Results

In each simulation, we constructed an environment with two tasks. For each, we sample 750 times from the first task, followed by 750 times from the second task. These 1,500 samples comprise the training data. We sample another 1,000 hold out samples to evaluate the algorithms. We fit a random forest (RF) (technically, an uncertainty forest which is an honest forest with a finite-sample correction (Mehta et al., 2019)) and a SynF. We repeat this process 30 times to obtain errorbars. Errorbars in all cases were negligible.

### F.1   Gaussian XOR

Gaussian XOR is two class classification problem with equal class priors. Conditioned on being in class 0, a sample is drawn from a mixture of two Gaussians with means $\pm \begin{bmatrix} 0.5, & 0.5 \end{bmatrix}^\mathsf{T}$, and variances proportional to the identity matrix. Conditioned on being in class 1, a sample is drawn from a mixture of two Gaussians with means $\pm \begin{bmatrix} 0.5, & -0.5 \end{bmatrix}^\mathsf{T}$, and variances proportional to the identity matrix. Gaussian XNOR is the same distribution as Gaussian XOR with the class labels flipped. Rotated XOR (R-XOR) rotates XOR by $\theta°$ degrees.

### F.2   Spirals

A description of the distributions for the two tasks is as follows: let $K$ be the number of classes and $S \sim$ multinomial$(\frac{1}{K}\vec{1}_K, n)$. Conditioned on $S$, each feature vector is parameterized by two variables, the radius $r$ and an angle $\theta$. For each sample, $r$ is sampled uniformly in $[0, 1]$. Conditioned on a particular class, the angles are evenly spaced between $\frac{4\pi(k-1)t_K}{K}$ and $\frac{4\pi(k)t_K}{K}$ where $t_K$ controls the number of turns in the spiral. To inject noise along the spiral, we add Gaussian noise to the evenly spaced angles $\theta' : \theta = \theta' + \mathcal{N}(0, \sigma_K^2)$.

Short-Time Fourier Transform Spectrogram of Number 5

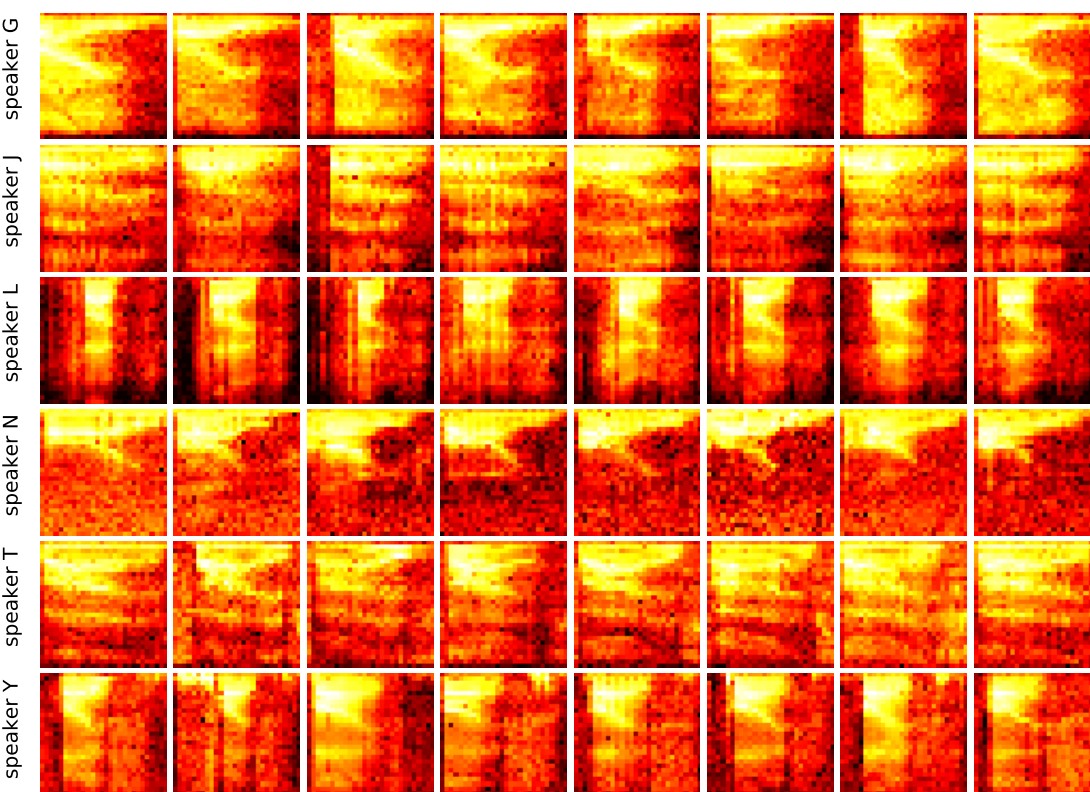

Figure 2: Spectrogram extracted from 8 different recordings of 6 speakers uttering the digit '5'.

The observed feature vector is then $(r\ \cos(\theta), r\ \sin(\theta))$. In Figure 1 we set $t_3 = 2.5$, $t_5 = 3.5$, $\sigma_3^2 = 3$ and $\sigma_5^2 = 1.876$.

Consider an environment with a three spiral and five spiral task (Figure 1). In this environment, axis-aligned splits are inefficient, because the optimal partitions are better approximated by irregular polytopes than by the orthotopes provided by axis-aligned splits. The three spiral data helps the five spiral performance because the optimal partitioning for these two tasks is relatively similar to one another, as indicated by positive forward transfer. This is despite the fact that the five spiral task requires more fine partitioning than the three spiral task. Because SynF grows relatively deep trees, it over-partitions space, thereby rendering tasks with more coarse optimal decision boundaries useful for tasks with more fine optimal decision boundaries. The five spiral data also improves the three spiral performance.

## G   Real Data Extended Results

### G.1   Spoken Digit Experiment

In this experiment, we used the spoken digit dataset provided in `https://github.com/Jakobovski/free-spoken-digit-dataset`. The dataset contains audio recordings from 6 different speakers with 50 recordings for each digit per speaker (3000 recordings in total). The experiment was set up with 6 tasks where each task contains recordings from only one speaker. For each recording, a spectrogram was extracted

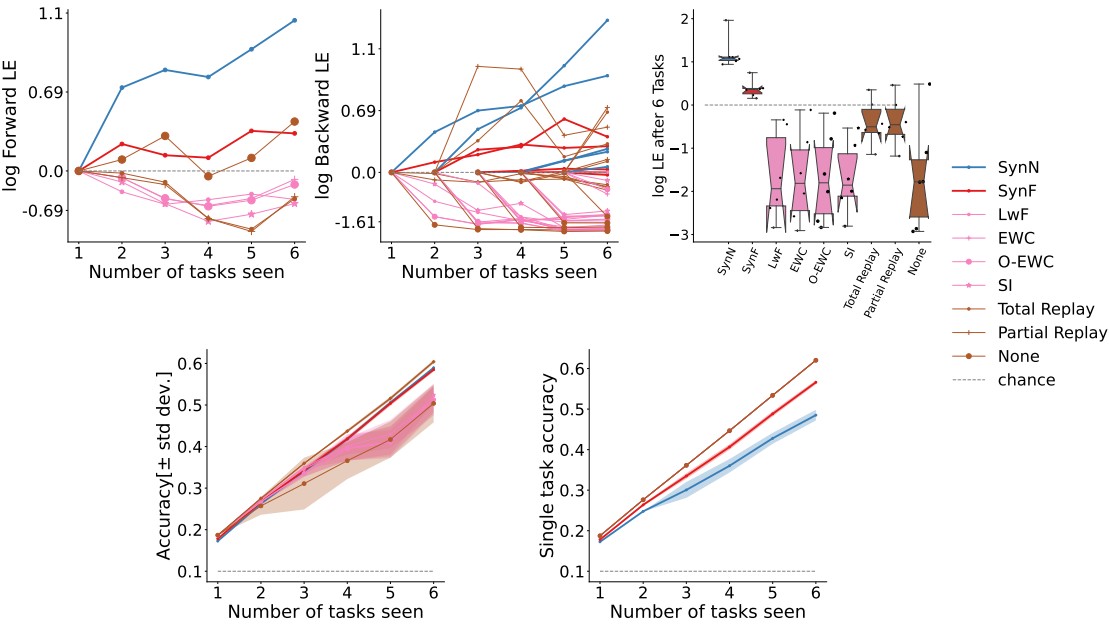

Figure 3: Both SynF and SynN show positive forward and backward transfer as well as synergistic learning for the spoken digit tasks, in contrast to other methods, some of which show only forward transfer, others show only backward transfer, with none showing both, and some showing neither.

Table 3: Hyperparameters for SynF in spoken digit experiment.

| Hyperparameters | Value |
|---|---|
| n_estimators (275 training samples per task) | 10 |
| max_depth | 30 |
| max_samples (OOB split) | 0.67 |
| min_samples_leaf | 1 |

Table 4: Task splits for CIFAR 10x10.

| Task # | Image Classes |
|---|---|
| 1 | apple, aquarium fish, baby, bear, beaver, bed, bee, beetle, bicycle, bottle |
| 2 | bowl, boy, bridge, bus, butterfly, camel, can, castle, caterpillar |
| 3 | chair, chimpanzee, clock, cloud, cockroach, couch, crab, crocodile, cup, dinosaur |
| 4 | dolphin, elephant, flatfish, forest, fox, girl, hamster, house, kangaroo, keyboard |
| 5 | lamp, lawn mower, leopard, lion, lizard, lobster, man, maple tree, motor cycle, mountain |
| 6 | mouse, mushroom, oak tree, orange, orchid, otter, palm tree, pear, pickup truck, pine tree |
| 7 | plain, plate, poppy, porcupine, possum, rabbit, raccoon, ray, road, rocket |
| 8 | rose, sea, seal, shark, shrew, skunk, skyscraper, snail, snke, spider |
| 9 | squirrel, streetcar, sunflower, sweet pepper, table, tank, telephone, television, tiger, tractor |
| 10 | train, trout, tulip, turtle, wardrobe, whale, willow tree, wolf, woman, worm |

using Hanning windows of duration 16 ms with an overlap of 4 ms between the adjacent windows. The spectrograms were resized down to $28 \times 28$. The extracted spectrograms from 8 random recordings of '5' for 6 speakers are shown in Figure 2. For each Monte Carlo repetition of the experiment, spectrograms extracted for each task were randomly divided into 55% train and 45% test set. As shown in Figure 3, both SynF and SynN show positive transfer and synergistic learning between the spoken digit tasks, in contrast to other methods, some of which show only forkward transfer, others show only backward transfer, with none showing both, and some showing neither.

## G.2 CIFAR 10x10

Supplementary Table 4 shows the image classes associated with each task number. Supplementary Figure 4 is the same as Figure 3 but with 5,000 training samples per task, rather than 500. Notably, with 5,000 samples, replay methods and Model Zoo are able to transfer both forward and backward as well. However, note that although total replay outperforms both SynF and SynN with large sample sizes, it is not a *bona fide* lifelong learning algorithm, because it requires $n^2$ time. Moreover, the replay methods will eventually forget as more tasks are introduced because it will run out of capacity.

## G.3 CIFAR Label Shuffling

Supplementary Figure 5 shows the same result as the label shuffling from Figure 4, but with 5,000 samples per class. The results for SynN and SynF are qualitatively similar, in that they transfer backward. The replay methods are also able to transfer when using this larger number of samples, although with considerably higher computational cost.

## G.4 CIFAR 10x10 Repeated Classes

We also considered the setting where each task is defined by a random sampling of 10 out of 100 classes with replacement. This environment is designed to demonstrate the effect of tasks with shared subtasks, which is a common property of real world lifelong learning tasks. Supplementary Figure 6 shows transfer efficiency of SynF and SynN on Task 1.

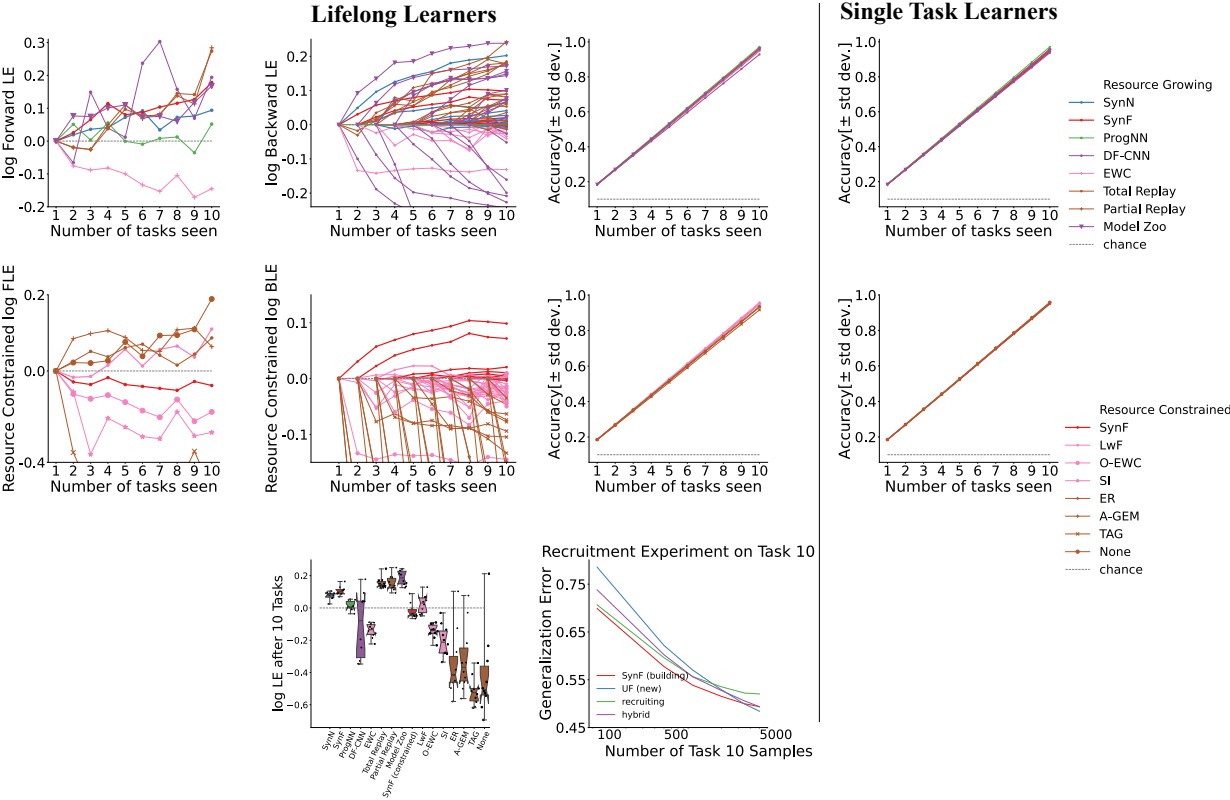

Figure 4: Performance of different algorithms on CIFAR 10x10 vision dataset for 5,000 training samples per task. SynN maintains approximately the same forward transfer (top left and middle left) and backward transfer (top and middle row second column) efficiency as those for 500 samples per task whereas other algorithms show reduced or nearly unchanged transfer. SynF still demonstrates positive forward, backward, and final transfer, unlike most of the state-of-the-art algorithms, which demonstrate forgetting. The replay methods, however, do demonstrate transfer, albeit with significantly higher computational cost.

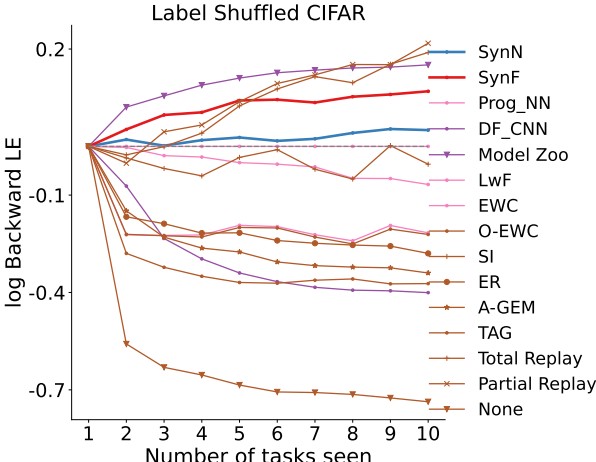

Figure 5: Label shuffle experiment on CIFAR 10x10 vision dataset for 5,000 training samples per task. Shuffling class labels within tasks two through nine with 5000 samples each demonstrates both SynF and SynN can still achieve positive backward transfer, and that the other algorithms that do not replay the previous task data fail to transfer.

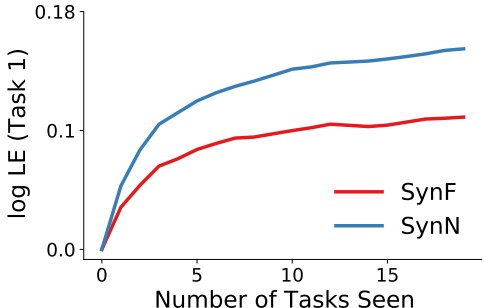

Figure 6: SynF and SynN transfer knowledge effectively when tasks share common classes. Each task is a random selection of 10 out of the 100 CIFAR-100 classes. Both SynF and SynN demonstrate monotonically increasing transfer efficiency for up to 20 tasks.

Table 5: 5-dataset details.

|  | Training samples | Testing samples |
|---|---|---|
| CIFAR-10 | 50000 | 10000 |
| MNIST | 60000 | 10000 |
| SVHN | 73257 | 26032 |
| notMNSIT | 16853 | 1873 |
| Fashion-MNIST | 60000 | 10000 |

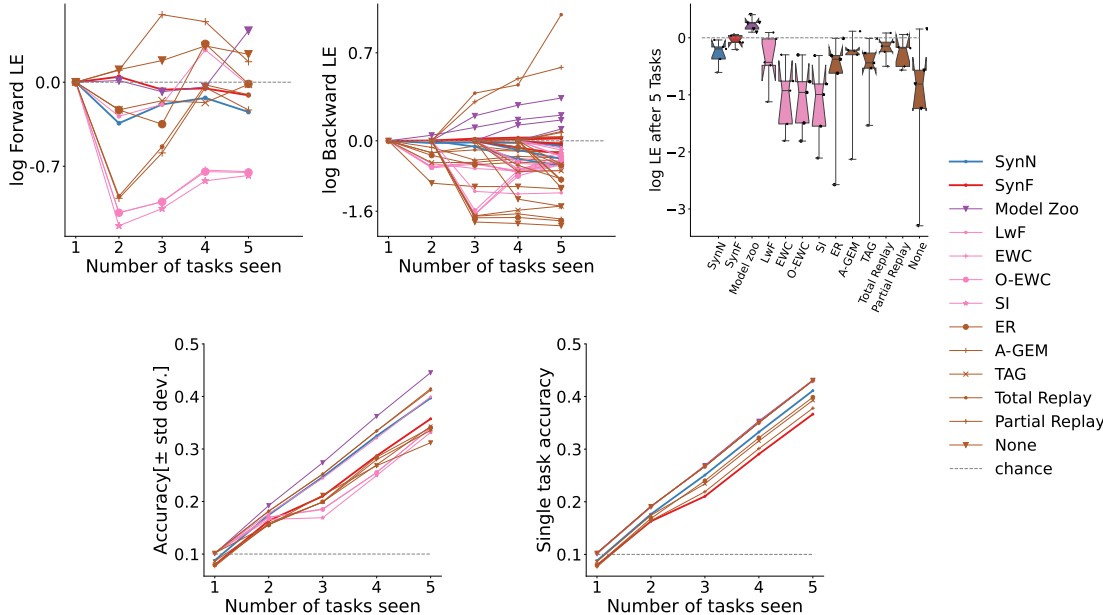

Figure 7: Performance of different lifelong learning algorithms on 5-dataset tasks.

### G.5 Five Dataset

In this experiment, we have used the 5-dataset provided in `https://github.com/pranshu28/TAG`. It consists of 5 tasks from five different dataset- CIFAR-10 (Krizhevsky, 2012), MNIST, SVHN (Netzer et al., 2011), notMNIST (Bulatov, 2011), Fashion-MNIST (Xiao et al., 2017). All the monochromatic images are converted to RGB format depending on the dataset. All images are then resized to $3 \times 32 \times 32$. As shown in table 5, training samples per task in 5-dataset is relatively higher than that of low data regime ideally considered in lifelong learning setting. However, SynN and SynFshow less forgetting than most of the benchmarking algorithms. On the other hand, model zoo shows comparatively better performance in higher task data setup.

### G.6 Split Mini-Imagenet

In this experiment, we have used the mini-imagenet dataset provided in `https://www.kaggle.com/datasets/whitemoon/miniimagenet`. The dataset was split into 20 tasks each 5 each. Each task has 2400 training samples and 600 testing samples. In this case, we positive FLE and BLE for both SynN and SynF. However, model zoo outperforms all the algorithms in this experiment.

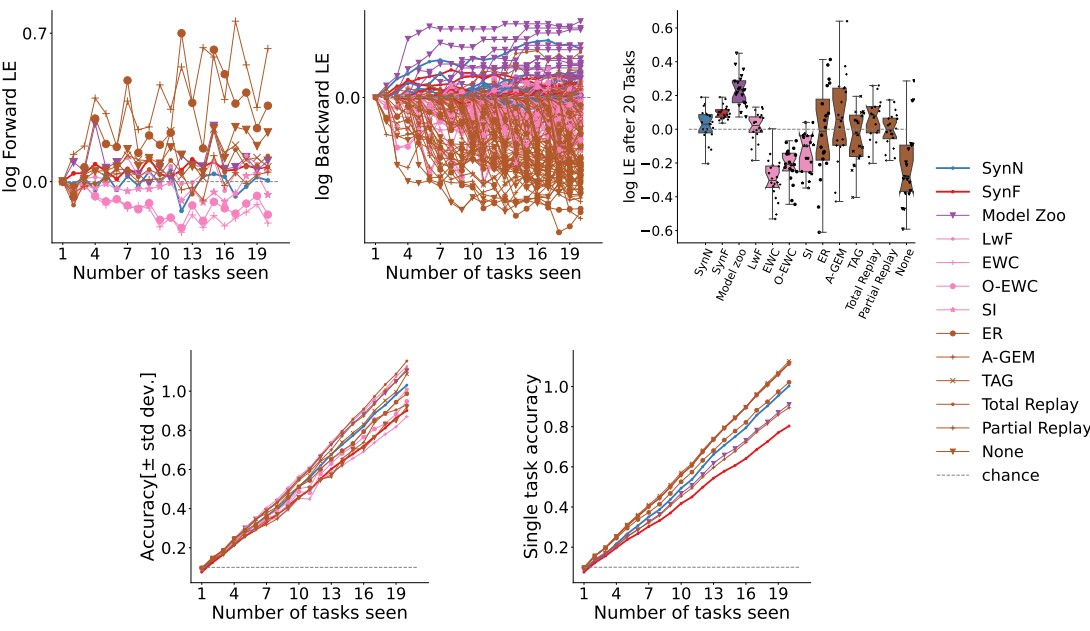

Figure 8: Performance of different lifelong learning algorithms on split mini-imagenet tasks.

