# OpenReview forum: "Representation Ensembling for Synergistic Lifelong Learning with Quasilinear Complexity"
_TMLR — Rejected by TMLR_

### Review · Reviewer_TiCt · 2022-07-11

**Summary Of Contributions:**

I have to confess that after reading the current paper in detail (in particular Section 3 several times) I still do not fully understand what's the problem setting studied in the paper. Different from the traditional lifelong learning setting, where at each time step the learning will face a new task / data distribution, from the notation introduced in Section 3, it seems that the same task $t$ could appear in multiple time steps in the sequence. This is different from the standard lifelong learning setting, but there is no motivation or justification on why this scenario is important, and what's the unique challenges in this new setting compared to the more standard one. There are other concerns in introducing the key metrics in Section 3 (see more detailed comments below), which prevents me from understanding and appreciating the contributions of this work.

Overall I suggest the authors perform a complete overhaul of the paper to make sure it is mathematically clear for both the problem setting as well as the proposed algorithm. In its current version, both parts to too vague for me to make a clear and informed decision. So I vote for rejection.

Detailed comments:
-   The presentation of the paper could be significantly improved. In particular, please put the citation inside a bracket (use \citep in Latex). The current presentation makes it very hard to parse and understand.

-   Section 2.2 could be more precise: please use a formula to summarize the loss function considered in the lifelong learning setting in this work. The current description seems a bit vague to me, i.e., is the goal to achieve low generalization error on the current task, or previous tasks. If the latter, because there are multiple previous tasks, which previous task is being considered specifically?

-   At the end of page 3, could you elaborate what does it mean for $S_n^t$ to be associated with task $t$? Is it a subset of data from $S_n$? If yes, what's the size of it?

-   Definition 1 seems very counter-intuitive to me. If my understanding is correct, $S_n^t$ is a subset of $S_n$, then likely the risk of the classifier on task $t$ obtained from using $S_n^t$ will be naturally larger than that of $S_n$, hence any trivial algorithm will be called "having learned all the tasks up to $t$". Please do correct me if this is not the case. The same comment also applies to the definitions of FLE abd BLE as well.

-   The main section, Section 4, is very unclear. The authors often mix the description of the model and results in the experiments, which IMO is a bad practice. Please first clearly lay out the proposed model, discuss its novelty by contrasting it to existing models, and then completely discuss experimental results in another section.


Minor comments:
-   I think the claim in the abstract that "classic machine learning ... using data only for the single task at hand" is over-simplifying and potentially misleading. There are many paradigms in ML (beyond lifelong learning) that also deals with more than one task, e.g., multitask learning, meta-learning, etc. Please be more precise.

-   Section 4, page 5, $h(\cdot) = w\circ v\circ u(\cdot)$: what on earth does this function decomposition have to do with information-theoretic hypothesis decomposition? To be honest I do not even know what is an information-theoretic hypothesis decomposition.

-   Section 4, page 5, posterior distribution: this is simply a distribution over the output space $\mathcal{Y}$, and it is not called a posterior distribution, which is a term often used in Bayesian approach.



**Requested Changes:**

Please see the comments above.

**Strengths And Weaknesses:**

Please see the comments above.

---

> ### Author Response · Authors · 2022-08-12
> **Authors' reply to Reviewer TiCt**
>
> We thank the reviewer for taking time to review our paper. We are sorry that the reviewer had to go through such grief while reviewing the paper. In what follows we have tried to address the concerns and we hope that it will clarify things further.
> - The purpose of lifelong learning is to mimic biological learning. As a biological learning agent can face the same task several times throughout its lifetime, a lifelong learner can do the same. Please see figure 1 in  ‘New, Alexander, et al. "Lifelong learning metrics." arXiv preprint arXiv:2201.08278 (2022)’. Again, to achieve backward transfer, we need to improve on the past tasks by virtue of seeing the current task. The purpose of trying to maintain backward learning efficiency is to do well whenever the agent sees repeated tasks from the past. Therefore, it is implicit in the definition of backward transfer that the same tasks can reappear.
>
> "The presentation of the paper could be significantly improved. In particular, please put the citation inside a bracket (use \citep in Latex). The current presentation makes it very hard to parse and understand."
> - We have fixed the citation style in the revised manuscript.
>
> "Section 2.2 could be more precise: please use a formula to summarize the loss function considered in the lifelong learning setting in this work. The current description seems a bit vague to me, i.e., is the goal to achieve low generalization error on the current task, or previous tasks. If the latter, because there are multiple previous tasks, which previous task is being considered specifically?"
> - We thank the reviewer for this nice suggestion. We have rephrased section 2.2 and used a formula to summarize the loss function in the revised draft. The goal of lifelong learning is to minimize the risk for each individual task seen so far by the agent as described in section 2.2. We are considering all the tasks seen so far, not a few of them.
>
> "At the end of page 3, could you elaborate what does it mean for $S_n^t$ to be associated with task t? Is it a subset of data from $S_n$? If yes, what's the size of it? "
> - We thank the reviewer for pointing out an important point in the text. We have updated the description at the end of page 3.
>
> "Definition 1 seems very counter-intuitive to me. If my understanding is correct, $S_n^t$ is a subset of $S_n$, then likely the risk of the classifier on task t obtained from using $S_n^t$ will be naturally larger than that of $S_n$, hence any trivial algorithm will be called "having learned all the tasks up to t". Please do correct me if this is not the case. The same comment also applies to the definitions of FLE and BLE as well."
> - We agree that $S_n^t$ is a subset of $S_n$. However, all of the task data are not available at the same time; rather they are introduced sequentially and there are distributional shifts among the tasks. Therefore, with computational and space constraints, it is not obvious for a classifier that the risk of the classifier on task t obtained from using $S_n^t$ will be naturally larger than that of $S_n$. Moreover, higher risk on the previous tasks while learning a new task (popularly known as forgetting) has bugged the AI community for a long time (zenke et al. 2017).
>
> "The main section, Section 4, is very unclear. The authors often mix the description of the model and results in the experiments, which IMO is a bad practice. Please first clearly lay out the proposed model, discuss its novelty by contrasting it to existing models, and then completely discuss experimental results in another section."
> - We have removed the results related texts from section 4.2 and put it in the beginning of section 5.2.
>
> "I think the claim in the abstract that "classic machine learning ... using data only for the single task at hand" is over-simplifying and potentially misleading. There are many paradigms in ML (beyond lifelong learning) that also deals with more than one task, e.g., multitask learning, meta-learning, etc. Please be more precise. "
> - This is a good point. We have rephrased the abstract in the revised draft.
>
> "Section 4, page 5, h(⋅)=w∘v∘u(⋅): what on earth does this function decomposition have to do with information-theoretic ..."
> - We apologize for the inconvenience faced by the reviewer with the term “information-theoretic hypothesis decomposition”. Encoder, channel and decoder are most widely known in information theory. We drew analogies among these terms and our hypothesis decomposition. However, we have rephrased the sentence to better reflect our intuition.
>
> "Section 4, page 5, posterior distribution: this is simply a distribution over the output space Y ..."
> - Thank you for the thoughtful feedback. We have rephrased the corresponding section to better reflect on our intuition. Moreover, please note that distribution over the output of deep-net models are treated as posteriors in the literature (Wenger, J. et al. 2020 “Non-parametric calibration for classification”).

---

### Review · Reviewer_UEUf · 2022-07-12

**Summary Of Contributions:**

This paper proposes a representation ensembling strategy for lifelong learning. According to their claim, the proposed "Synergistic Forests" and "Synergistic Networks" methods are the only methods that care about both forward and backward transfer among compared methods. To show the effectiveness of the proposed methods, they also propose a metric called "learning efficiency." Experimental results on the synthetic Gaussian XOR problem, its variants, and CIFAR-10x10 (and spoken digit dataset in Appendix F) show the effectiveness of the proposed methods in terms of the proposed learning efficiency metric.

**Broader Impact Concerns:**

Nothing special.

**Requested Changes:**

Please address concerns above. In particular, I'd like to see the absolute performance of the proposed method. Or, reporting results using the metrics in Lopez-Paz & Ranzato (2017) and/or Diaz-Rodriguez et al. (2018) should be better.

**Strengths And Weaknesses:**

Strengths:

+ This work studies they way to extend lifelong learning models, which I believe should be interesting to TMLR's audiences, while not studied so frequently.

Weakness:

- This work is not supported by "accurate, convincing and clear evidence," which is one of the most important criteria when making decisions in TMLR. The proposed evaluation metric is not intuitive, and the absolute performance of the proposed method is not presented. For example, I think a model making random decisions will have a better performance than all compared methods (other than the proposed methods) in the proposed metric, and a model that learns very slowly will have a good performance, as slow learning cannot be discriminated with positive forward/backward transfer. To avoid this criticism, you need to report the absolute performance like top-1 (average) accuracy, as many other continual learning works do.

Major issues:

- The intuition behind the proposed evaluation criterion "learning efficiency" is not clear. Could you elaborate more on the learning efficiency? Why do you say "the algorithm f has learned all tasks" if the risk computed only on task t data is larger than the one computed on all data?

- Metrics to measure the forward and backward transfer have already been proposed in Lopez-Paz & Ranzato (2017) (and I believe these metrics have been widely used in continual learning literature), and more metrics are also suggested by Diaz-Rodriguez et al. (2018), where some of them might be useful to measure the increasing model size for this paper. What are the problems in the metrics proposed in these works?

(Diaz-Rodriguez et al., 2018) Don't forget, there is more than forgetting: new metrics for Continual Learning. arXiv:1810.13166.

- Why do you care about the task-aware setting only? Does it mean that the proposed methods are not extendable to the task-agnostic setting?

- When ensembling representations, the information from other encoders can be either redundant (independent to the current task) or abused as a shortcut (such that it leads to overfitting to the current task). How do you guarantee that the model does not suffer from these issues?

- "The channels are learned via k-Nearest Neighbors (k-NN) Stone (1977)" in Section 4.2 sounds not practical, as all training data should be memorized to perform k-NN.

- Regarding "SynN excludes those lateral connections, thereby greatly reducing the number of parameters and train time" in Section 4.2, how does it specifically reduce the number of parameters? Removing connections does not mean directly to reduce the number of parameters.

- Regarding "SynN leverages representations from all J tasks. This difference enables backward transfer" in Section 4.2, how does SynN exactly leverage representations from all J tasks and enables backward transfer? I think that is one of the most interesting property of the proposed method, but I could not find the details and analyses on it.

- The 8-layer CNN architecture is too simple, so the results might not be scalable to modern deep architectures.


Minor issues:

- The citation format is problematic. You should cite a paper in the format of "(A, 2022)" instead of "A (2022)" if it is an adverb.

- Typo: "related to his -> related to this" in the first paragraph of Section 3.

- Typo: "The channels ensembles -> The channels ensemble" in the third paragraph of Section 4.

- Typo: there are inconsistent terms, like semiparametric vs. semi-parametric, nonparametric vs. non-parametric.

- Drawing multiple curves for the same method looks not a good way of visualizing results. You may want to average the numbers and show one curve per method.

- Navigating the pdf file via adobe acrobat reader is laggy. You may want to carefully check if there is an issue when compiling tex files.

---

> ### Author Response · Authors · 2022-08-12
> **Authors' reply to Reviewer UEUf**
>
> We thank the reviewer for the insightful comments that have enhanced our work.
> - Getting higher transfer learning by virtue of seeing additional tasks may be harder to get at 90% accuracy level than that of at chance level.  We ensured the benchmarking algorithms start from similar accuracies. We have included a global accuracy plot after each task (a figure similar to the one in Fig. 1 b) provided in Diaz-Rodriguez et al., 2018. We have also added the performance of the chance algorithm in Fig 3 and edited the text in the beginning of page 11.
> - We consider two generalization errors for the same model namely, errors when it has access to that particular task training data only and when it has access to all the training data up to that task. If the model has successfully leveraged the data from the other tasks, then it will have less generalization error compared to the case when it has only access to a single task data. Therefore, we say that a model has successfully transferred knowledge from the other task data to the current task t if it results in less generalization error of the model on that particular task, i.e., $LE_n^t(f) > 1$. This transfer of knowledge can come both from all the past tasks (forward transfer) and from the future tasks (backward transfer).
> - The proposed metrics in Diaz-Rodriguez et al., 2018 consider the difference of accuracy of the model at different stages of learning and in a similar way we consider the difference of log generalization error of the model at different stages of learning. However, averaging over tasks disregards information regarding the dynamics of the model after each task. For example, if a model performs exceptionally well on a particular task it will mask the poor performance it may have on other tasks. Again, while measuring forward transfer we consider the improvement in the performance of the model by virtue of seeing all the tasks data so far compared to the one where the model has access to a specific task data. In equation 4 of the suggested paper, the accuracy should be compared with respect to the chance accuracy. Otherwise, for a model operating at random, two different tasks with two different chance accuracy may result in positive forward transfer without meaningfully learning anything from other task data. We also have LE = FLE $\times$ BLE.
> - We consider task-aware settings for simplicity. A probable task-agnostic solution for the existing approach can be proposed by training a model to detect the task and learn the probability P(t| X=x). Thereafter, we can get the task independent posterior as-
> $P(Y=y|X=x) = \sum _t P(Y=y|X=x, T=t) P(T=t|X=x).$
> Then the predicted class label will be argmax over all the labels encountered by the model so far. However, verifying the model on different task-agnostic scenarios requires more work.
> - We agree with the reviewer that all of the encoders learned from all the tasks may not be helpful to the current task.These tasks are popularly known as adversarial task in the literature. We have demonstrated the effect of adversarial tasks (RXOR simulation, label shuffle and rotated cifar experiment) in the draft(section 5.2.2., 5.3.4).
> - In our present implementation, we need to save all the training data. The space complexity for a random forest is O(Tdn log^2(n)) and therefore, saving the training data does not add to the space complexity of the proposed approach. However, saving data can be avoided by learning a generative model simultaneously.
> - If we understand the question, then the answer is- in Prog-NN (Rusu et al., 2016), there are lateral connections between the task-dependent column from each layer of the past column to each layer of the next column. These connections have weight parameters that contribute to increasing the total parameters of the model with additional tasks.
> - In SynN, all of the representations learned from J tasks interact with each other through the channel layer (KNN and posterior averaging).
> - The main point of the paper is to demonstrate potential efficacy of ensembling representations. However, in this paper, we have used the same architecture as used in Van de Ven GM et al., 2020 to demonstrate different aspects of continual learning . As described in the last paragraph of section 4, our proposed approach is based on partition and vote (Priebe, Carey E., et al. 2020). Random forest and ReLU-net partitions the feature space into a set of affine polytopes. Therefore, each encoder represents a set of polytopes learned on a particular task. Next while inferring on a particular task, we push the corresponding task training data through all the polytopes in all of the encoders. Each set of polytopes gives a different estimate of the posterior around the inference point and the channel averages over them producing a better estimate of the posterior (reduces variance of the posterior) compared to that of the single task estimator improving performance using multiple task data.

---

> > ### Author Response · Authors · 2022-08-12
> > **Minor issues**
> >
> > Minor issues:
> > The citation format is problematic. You should cite a paper in the format of "(A, 2022)" instead of "A (2022)" if it is an adverb.
> > Typo: "related to his -> related to this" in the first paragraph of Section 3.
> > Typo: "The channels ensembles -> The channels ensemble" in the third paragraph of Section 4.
> > Typo: there are inconsistent terms, like semiparametric vs. semi-parametric, nonparametric vs. non-parametric.
> > - We have fixed the citation style and the typos.
> >
> > "Drawing multiple curves for the same method looks not a good way of visualizing results. You may want to average the numbers and show one curve per method."
> > - We thank the reviewer for pointing it out. We understand that we did not do a good job in explaining our results. Our proposed performance metrics require multiple curves to give a detailed idea about the performance. We have tried to improve the description in section 5.3 second paragraph. We wanted to demonstrate that our proposed approach (red and blue curves) remains above other curves most of the time showing superiority of our approaches as other approaches hardly show any positive transfer.
> >
> > "Navigating the pdf file via adobe acrobat reader is laggy. You may want to carefully check if there is an issue when compiling tex files."
> > - We have fixed all the latex warnings. Do you still have issues while navigating across the pdf?

---

### Review · Reviewer_9sGP · 2022-08-05

**Summary Of Contributions:**

This paper introduces a new algorithm for lifelong learning that ensembles the representation from both previous and future tasks to make predictions and hence has the potential to enable both forward and backward transfer. The paper also introduces new metrics to measure forward and backward transfer. Experiments on simple XOR and CIFAR tasks show that the proposed method has better transfer.

**Broader Impact Concerns:**

None.

**Requested Changes:**

Major comments:

1. It is difficult to implement the algorithm based on the description in the paper. Even though there is some pseudo-code in the appendix, it is only for the decision tree. For the neural network version, it is not clear how the channel is implemented and how the decoder is implemented. Authors need to clearly explain their algorithm, provide proper pseudo code and also code to reproduce their results.

2. In the SynF algorithm, wouldn't different trees produce different length one-hot encoding? It is not clear to me how these different length one-hot encoding are combined to make a prediction.

3.  About the line: “Interestingly, when using 5,000 samples per task, replay methods are able to demonstrate positive forward and backward transfer (Supplementary Figure 4), although they require quadratic time.” The time taken by the replay based methods depends on how frequent you replay. There are many recent work which shows that you can replay less frequently and get most of the benefits on the replay in which case it is more close to linear time than quadratic time.

4. The paper mentions in few places that SynF and SynN store old data. I don’t see how this old data is used by the algorithm. Can you please explain?

5. Several important references are missing.

https://arxiv.org/abs/1902.10486 - This paper is still the state-of-the-art replay method for lifelong learning. However, there is no mention of this work. A comparison with this work is essential.

https://openreview.net/forum?id=Hkf2_sC5FX - A GEM is one of the most popular methods for lifelong learning. A comparison with this work is essential.

https://arxiv.org/abs/2106.03027 - This work also proposes an ensemble-based method for lifelong learning. A comparison with this work is essential.

https://arxiv.org/abs/1811.07017 - This paper talks about how you can avoid quadratic growth in the number of parameters by using network expansion techniques sparsely. A comparison is not needed. But it is worth mentioning in the related work.

https://arxiv.org/abs/2105.05155 - This paper proposes an optimization-based solution for lifelong learning that promotes knowledge transfer. A comparison with this work is essential.

6. Experiments are one of the most important weaknesses of this paper. All the experiments on based on simple XOR task and simple CIFAR-100 task. Authors should do split-mini-imagenet, split-CUB, and 5-dataset tasks used in many papers including https://arxiv.org/abs/2105.05155

7. It is very important that authors report all the traditional metrics for forgetting, forward transfer, and backward transfer and also show empirically why the new metrics are needed. When comparing with existing methods using your own metric, it is not clear if the performance gain from previous methods is really what is claimed in the paper.

Minor comment:

1. Please use \citep whenever you want to refer the paper all in bracket. For example, the first line is:

Learning is the process by which an intelligent system improves performance on a given task by leveraging data Mitchell (1999).

However, it should be:

Learning is the process by which an intelligent system improves performance on a given task by leveraging data (Mitchell 1999).


2. In the line “Lifelong learning approaches can be divided into those with fixed computational space resources, and those with growing space resources (which we refer to as ‘fixed resources’ hereafter).” you refer to growing space resources as ‘fixed resources’. Is this a typo?


**Strengths And Weaknesses:**

Strengths:

1. The proposed algorithm is very interesting.
2. The focus is more on transfer than on forgetting which is currently needed in the lifelong learning community.

Weakness:

1. Experiments are very toyish and more experiments are needed to understand the true performance of the algorithm.
2. Some important baselines missing.
3. When introducing both a new algorithm and a new metric, authors should also do a lot of experiments to justify how the new metric is interesting when compared to existing metrics. This is currently missing.

---

### Decision · Action_Editors · 2022-10-06

**Recommendation:** Reject

**Comment:**

The reviewers found the proposed algorithm would be of interest to the TMLR audience. However, they noted that the experiments were limited and that the writing could be improved. In response to this, the authors added more complex datasets and metric justifications in the appendix. These are important improvements and as such they should be integrated into the main paper so that it is more complete by itself. I recommend for the authors to further improve the writing of the paper as the reviewers suggested.

**Audience:**

The paper is relevant to TMLR's audience since lifelong learning remains an important open problem in the field.

**Claims And Evidence:**

This paper proposes a new approach to lifelong learning using representation ensembles. The paper provides empirical evidence including on CIFAR. Later experiments were added to the appendix in response to the reviewers' comments.